# Dynamic influences on static measures of metacognition

Kobe Desender [1,2,3 ✉], Luc Vermeylen [3] & Tom Verguts [3]

Humans differ in their capability to judge choice accuracy via confidence judgments. Popular signal detection theoretic measures of metacognition, such as M-ratio, do not consider the dynamics of decision making. This can be problematic if response caution is shifted to alter the tradeoff between speed and accuracy. Such shifts could induce unaccounted-for sources of variation in the assessment of metacognition. Instead, evidence accumulation frameworks consider decision making, including the computation of confidence, as a dynamic process unfolding over time. Using simulations, we show a relation between response caution and M-ratio. We then show the same pattern in human participants explicitly instructed to focus on speed or accuracy. Finally, this association between M-ratio and response caution is also present across four datasets without any reference towards speed. In contrast, when data are analyzed with a dynamic measure of metacognition, v-ratio, there is no effect of speed-accuracy tradeoff.

[1] Brain and Cognition, KU Leuven, Belgium. [2] Department of Neurophysiology and Pathophysiology, University Medical Center Hamburg-Eppendorf, Hamburg, Germany. [3] Department of Experimental Psychology, Ghent University, Ghent, Belgium. ✉email: Kobe.Desender@kuleuven.be

When asked to explicitly report how sure they are about their decisions, humans often claim high confidence for correct and low confidence for incorrect decisions. This capacity to evaluate the accuracy of decisions is often referred to as metacognitive accuracy. Although metacognitive accuracy about perceptual decisions is generally high[1], it varies significantly between participants[2] and between conditions[3]. Such differences in metacognitive accuracy are associated with important real-life consequences, as they relate, for example, to political extremism[4] and psychiatric symptoms[5]. Moreover, an increasing number of researchers across different fields are starting to investigate to what extent observers can evaluate their own performance in different domains of cognition, such as sensorimotor uncertainty[6], motor movements and imagery[7], metric error monitoring[8], value-based decisions[9], group-decision making[10] and probabilistic learning[11].

A debated question is how to quantify metacognitive accuracy. One prominent issue why one cannot simply calculate the correlation between confidence and choice accuracy[12] is that this confounds choice accuracy with metacognitive accuracy; i.e. it is much easier to detect one's own mistakes in an easy task than in a hard task. Different solutions have been proposed in the literature, such as using coefficients from a logistic mixed-model[13], type 2 ROC curves[2], and meta-d'[14,15]. Rather than providing an in-depth discussion and comparison of these different measures, we here focus on one prominent static approach, namely the meta-d' framework, the state-of-the-art measure of metacognitive accuracy[16]. The meta-d' approach is embedded within signal detection theory, and quantifies the extent to which confidence ratings discriminate between correct and incorrect responses (meta-d') while controlling for first-order task performance (d'). Because both measures are on the same scale, one can calculate the ratio between both, meta-d'/d', also called M-ratio, often referred to as metacognitive *efficiency*. When M-ratio is 1, all available first-order information is used in the (second-order) confidence judgment. When M-ratio is smaller than 1, metacognitive sensitivity is suboptimal, meaning that not all available information from the first-order response is used in the metacognitive judgment[16]. This measure has been used to address a variety of questions, such as whether metacognition is a domain-general capacity[3,17,18], the neural correlates of metacognition[19–22], how bilinguals differ from monolinguals[23], and how individual differences in metacognitive accuracy correlate with various constructs[4,5].

An important limitation is that the meta-d' framework (just like the other static approaches cited above), does not consider dynamic aspects of decision making. Put simply, this measure depends on end-of-trial confidence and choice accuracy, but not on the response process governing the choice and its resulting reaction time. It is well known, however, that choice accuracy depends on response caution; i.e. choice accuracy decreases when participants are instructed to be fast rather than to be correct[24,25]. The fact that static approaches of metacognition do not consider response caution is problematic because it confounds ability with caution: when focusing on speed rather than accuracy, one will produce many errors due to premature responding, and those errors are much easier to detect compared to errors resulting from low signal quality[26]. Importantly, detecting "premature" errors does not imply "good metacognition" per se, but instead simply depends on one's level of response caution.

To account for dynamic influences on metacognition, we propose to instead quantify metacognitive accuracy in a dynamic framework[27,28]. Sequential sampling models explain human decision making as a dynamic process of evidence accumulation[29–31]. Specifically, decisions are conceptualized as resulting from the accumulation of noisy evidence towards one of two decision boundaries. The first boundary that is reached,

triggers its associated decision. The height of the decision boundary controls the response caution with which a decision is taken[24,25]. When lowering the boundary, decisions will be faster but less accurate; when increasing the boundary, decisions will be slower but more accurate. The prototypical dynamic sampling model is the drift diffusion model (DDM). In this model, confidence can be quantified as the level of evidence integrated after additional post-decisional evidence accumulation[27,28,32,33]. This formalization of confidence, visualized in Fig. 1A, can explain the typical finding that trials with strong evidence are more likely to be judged with high confidence than trials with weak evidence. As mentioned, the process of evidence accumulation terminates at the first boundary crossing. At that moment in time, given fixed decision boundaries, the level of evidence, $e_{t, X}$, is constant, where $e$ is the level of evidence at time $t$, $t$ is the timing of boundary crossing and $X$ is the choice made[27,32–34]. In typical experiments, however, confidence judgments are provided separately in time (at time $t + s$, i.e., in a separate judgment after the choice), allowing post-decisional evidence accumulation. As a consequence, confidence can be quantified as $e_{t+s, X}$[27,28,34]. This implies that a choice will be made once the integrated evidence reaches boundary $a$, but confidence is only computed after additional evidence accumulation (see Fig. 1A for illustration).

Within this formulation, good metacognitive accuracy can be considered as the ability to distinguish correct choices versus error choices based on $e_{t+s, X}$, i.e., based on confidence. Critically, the difference in the quantity $e_{t+s, X}$ for correct choices versus error choices, directly depends on the strength of post-decisional accumulation. This is visually depicted in Fig. 1B: when post-decisional evidence accumulation is only driven by noise (i.e., post-decision drift rate is zero), model predicted confidence will be identical for correct and error trials (Fig. 1B, left panel). On the other hand, when post-decisional evidence accumulation is very strong (i.e. post-decision drift rate is high) model predicted confidence strongly dissociates between corrects and errors, reflecting good metacognition (Fig. 1B, right panel). From the above, it becomes clear that we can use the strength of post-decisional evidence accumulation as a dynamic measure of metacognitive accuracy. For comparison with the M-ratio framework, we quantified v-ratio as the ratio between post-decision drift rate and (pre-decision) drift rate. Figure 1B shows post-decision accumulation for three scenarios with varying levels of v-ratio. As can be seen, if v-ratio is zero (left panel), additional evidence meanders adrift for both corrects and errors, and the model does not detect its own errors, i.e., representing a case of poor metacognitive accuracy. If, however, v-ratio equals 1 (i.e., post-decision drift rate and drift rate are the same), additional evidence confirms most of the correct choices (i.e., leading to high confidence) and disconfirms most of the error choices (i.e., leading to low confidence), i.e., representing good metacognitive accuracy. We thus propose that v-ratio can be used as a dynamic measure of metacognitive accuracy. In the current work, we show that decreased response caution is associated with increased estimates of M-ratio whereas v-ratio is independent of response caution. This is the case both in drift diffusion model simulations, in two experiments where participants are explicitly instructed to change the tradeoff between speed and accuracy, and in four datasets where no reference regarding speed or accuracy is provided.

## Results

### Model simulations reveal a link between response caution and M-ratio. We simulated data from a drift diffusion model with additional post-decisional evidence accumulation (see Fig. 1A). Decision confidence was quantified as the level of integrated

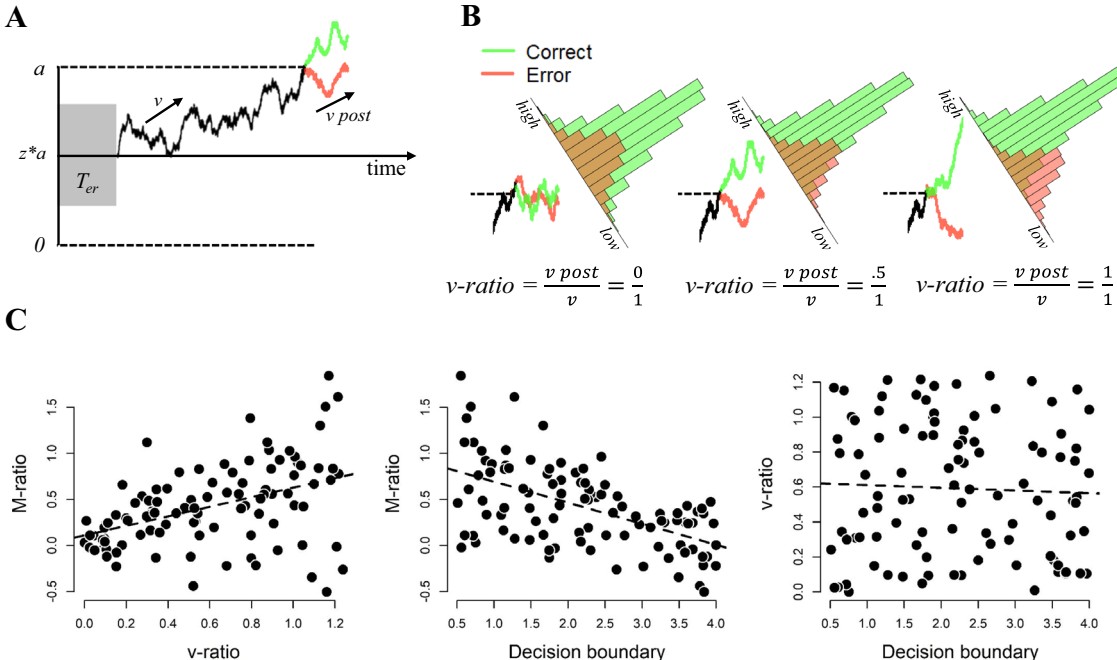

**Fig. 1 Quantifying metacognitive accuracy within an evidence accumulation framework. A** Noisy sensory evidence accumulates over time, until the integrated evidence reaches one of two decision boundaries (a or 0). After the decision boundary is reached, evidence continues to accumulate. Model confidence is quantified as the integrated evidence after post-decisional evidence accumulation. **B** Histograms of model-predicted confidence for different levels of v-ratio (reflecting the ratio between post-decision drift rate and drift rate). Higher levels of v-ratio are associated with better dissociating corrects from errors. **C** Simulations from this dynamic evidence accumulation model show that v-ratio captures variation in M-ratio ($r = 0.436$; left panel), and critically, that M-ratio is also related to the differences in decision boundary ($r = -0.552$; middle panel). By design, decision boundary and v-ratio are unrelated to each other ($r \sim 0$; right panel). Data are based on $N = 100$ simulations. Source data are provided as a Source Data file.

| | 1 | 2 | 3 | 4 | 5 |
|---|---|---|---|---|---|
| **Table 1 Correlation table of the parameters from the model simulation, based on $N = 100$ simulations.** | | | | | |
| 1. Drift rate | - | | | | |
| 2. Non-decision time | $r = -0.04, p = 0.696$ | - | | | |
| 3. Decision boundary | $r = -0.03, p = 0.769$ | $r = 0.02, p = 0.845$ | - | | |
| 4. v-ratio | $r = 0.02, p = 0.845$ | $r = -0.07, p = 0.493$ | $r = -0.04, p = 0.695$ | - | |
| 5. M-ratio | $r = -0.24, p = 0.017$ | $r = 0.01, p = 0.922$ | $r = -0.55, p < 0.001$ | $r = 0.44, p < 0.001$ | - |

evidence after additional post-decisional evidence accumulation[27,32–34]. We simulated data for 100 agents with 1000 observations each; for each agent, a different random value was selected for drift rate, non-decision time, decision boundary and post-decision drift rate. Importantly, we made sure that the simulated data showed the same correlation between average choice RTs and average confidence RTs as seen in the empirical data (see Methods). We then used these data to compute M-ratio, after dividing confidence ratings into four bins, separately for each observer (which is needed to compute meta-*d'*). As explained before, v-ratio was computed as the ratio between post-decision drift rate and drift rate. The results of our simulation study showed that, first, there was a clear positive relation between M-ratio and v-ratio, $r(98) = 0.436$, $p < 0.001$, reflecting that M-ratio captures individual variation in metacognition (Fig. 1C, left panel). However, we also observed a strongly negative relation between M-ratio and decision boundary, $r(98) = -0.552$, $p < 0.001$ (Fig. 1C, central panel). This shows that M-ratio is highly dependent on the speed-accuracy tradeoff that one adopts: Lower bounds are associated with higher M-ratio. Intuitively, this occurs because lowering the decision boundary increases the probability of premature errors due to noise in the accumulation process. Given that M-ratio reflects a ratio

between meta-*d'* and *d'*, increasing the probability of premature errors can affect M-ratio in two ways: first, a lower decision boundary decreases *d'*, and will therefore have the effect that it increases M-ratio. Second, it is known that premature errors are easier to detect[26], and therefore a lower decision boundary might increase meta-*d'*, and will therefore increase M-ratio. Finally, by design there was no relation between v-ratio and decision boundary, $r(98) = -0.044$, $p = 0.670$ (Fig. 1C, right panel). In the Supplementary Information, Supplementary Note 2, we report an additional analyses into the dynamics of v-ratio using type II area under the curve (AUC). The full correlation matrix is shown in Table 1.

**Experiment 1: Explicit speed-accuracy instructions affect static but not dynamic measures of confidence.** Next, we tested these model predictions in an experiment with human participants. We recruited 32 human participants who performed a task that has been widely used in the study of evidence accumulation models: discrimination of the net motion direction in dynamic random dot displays[29]. Participants were asked to decide whether a subset of dots was moving coherently towards the left or the right side of the screen (See Fig. 2A). The percentage of dots that coherently

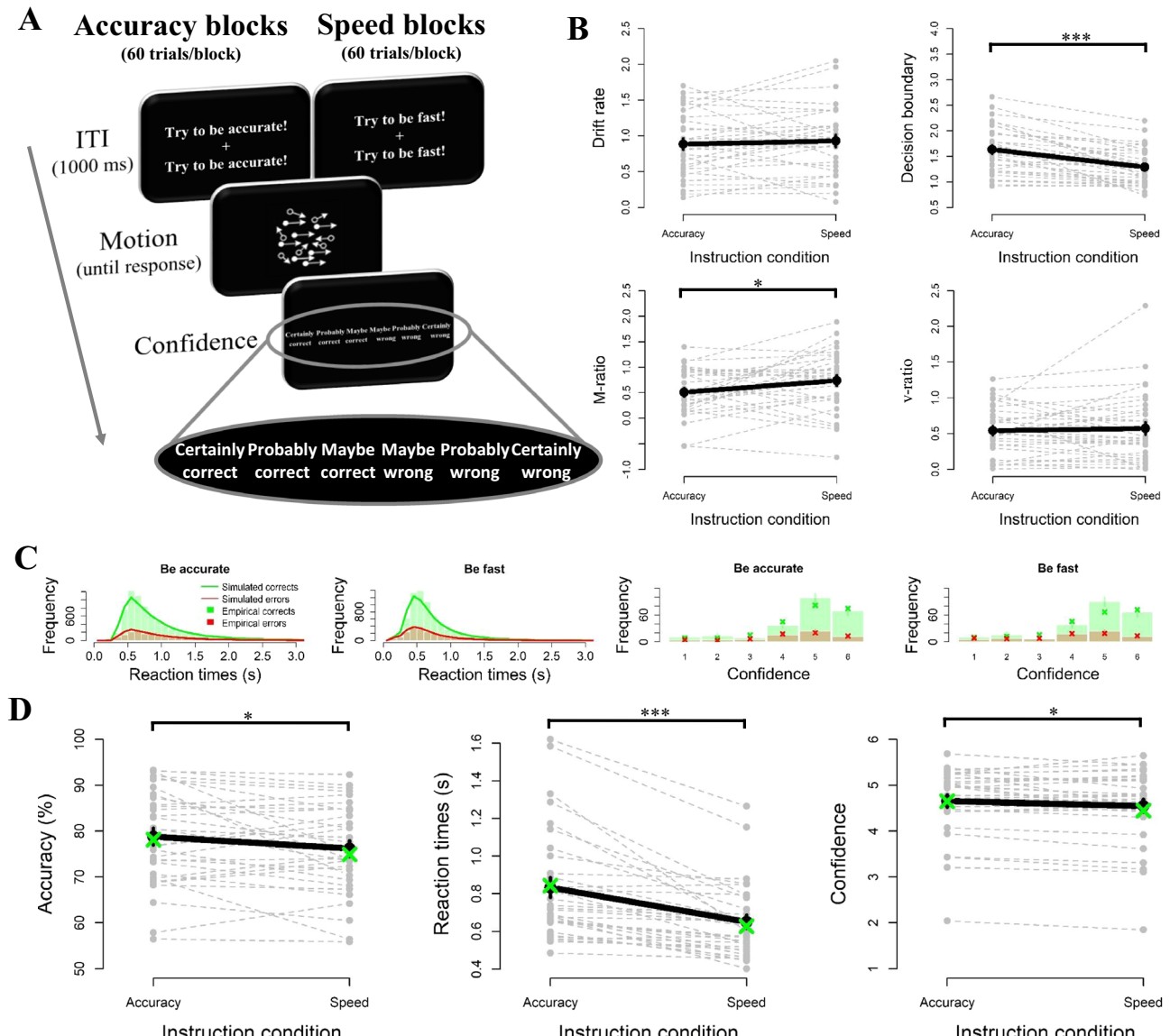

**Fig. 2 The influence of speed-accuracy instructions on metacognitive accuracy (Experiment 1). A** Sequence of events in the experimental task. Participants ($N = 32$) decided whether the majority of dots were moving left or right, by pressing "C" or "N" with the thumbs of both hands. Immediately after their choice, they then indicated their level of confidence using a six-point scale. Depending on the block, instructions during the ITI were either to focus on choice accuracy or to focus on speed. **B** Fitted parameters of a drift diffusion model with additional post-decision accumulation. Fitted decision boundaries were lower in the speed vs accuracy condition, $t(31) = 5.59$, $p < 0.001$, whereas drift rates did not differ, $p = 0.478$. Critically, M-ratio was higher in the speed vs accuracy condition, $t(31) = 2.29$, $p = 0.029$, whereas v-ratio did not differ between both instruction conditions, $p = 0.647$. **C** Distribution of reaction times and confidence for empirical data (bars) and model fits (lines or crosses), separately for corrects (green) and errors (red). **D** Participants were faster, $t(31) = 5.67$, $p < 0.001$, less accurate, $t(31) = 2.20$, $p = 0.035$, and less confident, $t(31) = 2.41$, $p = 0.022$, when instructed to focus on speed rather than on accuracy. Note: grey lines show individual data points; black lines show averages; green dots show model fits; error bars reflect SEM; ***$p < 0.001$, **$p < 0.01$, *$p < 0.05$. Source data are provided as a Source Data file.

moved towards the left or right side of the screen (controlling decision difficulty) was held constant throughout the experiment at 20%. After their choice, participants indicated their level of confidence by pressing one of six buttons. Critically, in each block participants either received the instruction to focus on choice accuracy ("try to decide as accurate as possible"), or to focus on speed ("try to decide as fast as possible"). Consistent with the instructions, participants were faster in the speed condition than in the accuracy condition, $M_{speed} = 650$ ms versus $M_{accuracy} = 832$ ms, $t(31) = 5.67$, $p < 0.001$, and more accurate in the accuracy condition than in the speed condition, $M_{accurate} = 78.8\%$ vs $M_{speed} = 76.2\%$, $t(31) = 2.20$, $p = 0.035$. To further corroborate

that indeed errors in the speed condition were mostly "fast" errors and errors in the accuracy condition were mostly "slow" errors, we divided each participant's error RTs into three equal-sized bins (fast, medium or slow; see Supplementary Fig. 1). We observed that in the fast bin there were more errors from the speed than from the accuracy condition ($M_{speed} = 30.3$ trials vs $M_{accuracy} = 13.5$ trials, $t(31) = -6.91$, $p < 0.001$), whereas in the slow bin the reverse was true ($M_{speed} = 15.0$ trials vs $M_{accuracy} = 29.4$ trials, $t(31) = 5.97$, $p < 0.001$). In the medium bin the number of trials did not differ between both instruction conditions ($M_{speed} = 24.0$ trials vs $M_{accuracy} = 20.0$ trials, $p = 0.074$). Note that median confidence RTs were different between the two

instruction conditions, $M_{speed} = 348$ ms versus $M_{accuracy} = 385$ ms, $t(31) = 2.32$, $p = 0.027$. There were no significant between-participants correlation between median choice RTs and median confidence RTs neither in the accuracy condition, $r(30) = .217$, $p = 0.232$, nor in the speed condition, $r(30) = -0.257$, $p = 0.156$. Finally, participants were more confident in the accuracy condition than in the speed condition, $M_{accuracy} = 4.65$ versus $M_{speed} = 4.55$, $t(31) = 2.41$, $p = 0.022$ (See Fig. 2D).

To shed further light on the underlying cognitive processes, we fitted these data using the evidence accumulation model described in Fig. 1A. The basic architecture of our model was a DDM, in which noisy perceptual evidence accumulates over time until a decision boundary is reached. Afterwards, evidence continued to accumulate to information confidence judgments. The distribution of post-decision evidence accumulation times was directly determined by the distribution of empirically observed confidence RTs[27]. In addition to drift rate, decision boundary and non-decision time, our model featured a free parameter controlling the strength of the post-decision evidence accumulation (v-ratio, reflecting the ratio between post-decision drift rate and drift rate) and two further parameters controlling the mapping from integrated evidence onto the confidence scale (see Methods). Generally, our model fitted the data well, as it captured the distributional properties of both reaction times and decision confidence (see Fig. 2C). As a first sanity check, we confirmed that decision boundaries were indeed different between the two instruction conditions, $M_{speed} = 1.29$ versus $M_{accuracy} = 1.63$, $t(31) = 5.59$, $p < 0.001$, suggesting that participants changed their decision boundaries as instructed. Also non-decision time tended to be a bit shorter in the speed condition compared to the accuracy condition, $M_{speed} = 306$ ms versus $M_{accuracy} = 349$ ms, $t(31) = 2.13$, $p = 0.041$. Drift rates did not differ between both instruction conditions, $p = 0.478$. There was a small but significant difference between the two instruction conditions for one of the two additional parameters controlling the idiosyncratic mapping between integrated evidence and the confidence scale (see Methods), reflecting that in the accuracy condition confidence judgments were slightly higher, $t(31) = 3.32$, $p = 0.001$, but not more variable, $t(31) = 1.72$, $p = 0.095$, compared to the speed condition.

We next focused on metacognitive accuracy in both conditions (see Fig. 2B). In line with the model simulations, our data showed that M-ratio was significantly affected by the speed-accuracy tradeoff instructions, $M_{speed} = 0.74$ versus $M_{accuracy} = 0.51$, $t(31) = 2.29$, $p = 0.029$. Consistent with the notion that metacognitive accuracy should not be affected by differences in decision boundary, v-ratio did not differ between both instruction conditions, $p = 0.647$. In the model just reported, all parameters were allowed to vary as a function of both instruction conditions. This allowed us to evaluate for each parameter whether or not it is affected by the instruction condition. Note, however, that qualitatively similar results were obtained when instead fixing non-decision time, $M$ and $SD$ across both conditions. Finally, in the Supplementary Information, Supplementary Note 1, we report a replication of Experiment 1 using a more fine-grained confidence scale, which yielded identical conclusions as the study reported here.

**Experiment 2A-D: Spontaneous differences in response caution relate to static but not dynamic measures of metacognitive accuracy.** Although Experiment 1 provides direct evidence that changes in decision boundary affect M-ratio, it remains unclear to what extent this is also an issue in experiments without speed stress. Notably, in many metacognition experiments, participants do not receive the instruction to respond as fast as possible.

Nevertheless, it remains possible that participants implicitly decide on a certain level of response caution. For example, a participant who is eager to finish the experiment quickly might adopt a lower decision boundary compared to a participant who is determined to perform the experiment as accurately as possible, thus leading to natural across-participant variation in decision boundaries. To evaluate this possibility, we examined the data of four experiments (equaling a total $N = 430$) in which participants did not receive any specific instructions concerning speed or choice accuracy. All experiments concerned a binary perceptual discrimination task with additional confidence ratings (see Methods). In Experiment 2 A ($N = 63$) and 2B ($N = 96$), participants were presented with two boxes filled with dots and had to decide which of the two boxes contained more dots by pressing the "S" or "L" key (corresponding to left and right, respectively). After their choice, participants used the same keys to move a cursor on a continuous confidence scale to indicate their level of confidence, and confirmed by pressing the enter key (Experiment 2 A); or they indicated their level of confidence by pressing one of six buttons at the top of their keyboard (Experiment 2B). In Experiment 2 C, participants ($N = 204$) were presented with two consecutive arrays of six Gabor patches, and were asked to decide in which of the two arrays one of the patches had a higher contrast. The response modalities were similar to those used in Experiment 2 A. Finally, in Experiment 2D participants ($N = 67$) decided whether the average color of eight elements was red or blue, by pressing the "C" or "N" key. Afterwards, they indicated their choice by pressing one of six buttons at the top of their keyboard.

The same evidence accumulation model as before was used to fit these data, and again this model captured both reaction times and decision confidence distributions for all datasets. We performed hierarchical mixed effects modeling to examine whether we could replicate the findings of Experiment 1, without an explicit manipulation of speed and accuracy, across the four datasets. Because two of the datasets contained very precise measurements of confidence RTs (Experiments 2B and 2D) and in two datasets confidence was provided by pressing cursors keys (Experiments 2 A and 2 C), we included confidence response mode as a factor in the model.

First, we tested whether M-ratio was predicted by the decision bound and whether this effect was independent from how confidence judgments were provided. To achieve this, we fitted the following hierarchical mixed effects model to the data:

$$\text{M-ratio} \sim \text{decision bound} * \text{confidence response mode} + (1|\text{experiment}) \tag{1}$$

where $(1|\text{experiment})$ reflects the hierarchical clustering of participants within experiments and * indicates that an interaction effect was estimated. Adding random slopes to the model did not increase model fit, so these were not included in the final model. As expected, we observed a significant negative relation between decision boundary and M-ratio, $b = -0.200$, $t(424) = -2.43$, $p = 0.015$ (see Fig. 3B). Importantly, this effect did not interact with confidence response mode, $p = 0.118$, nor was there a main effect of confidence response mode, $p > 0.337$. Thus, this analysis demonstrates that there was evidence across datasets for a relation between M-ratio and decision boundary irrespective of the way in which confidence was measured. This analysis again suggests that M-ratio is not a pure measure of metacognition, but is confounded with response caution.

Second, we addressed the question whether v-ratio is a good alternative measure of metacognition. To do so, we tested whether v-ratio relates to M-ratio, showing that both measures capture shared variance in metacognition, and whether v-ratio is unrelated to the decision boundary, showing that v-ratio is

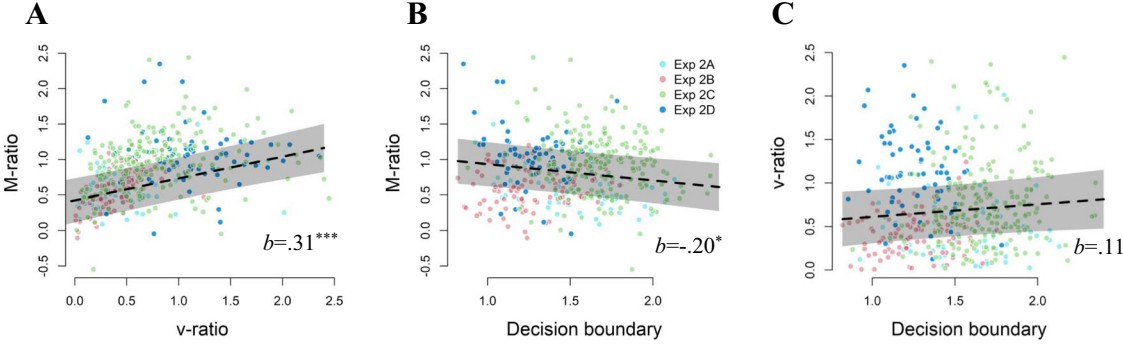

**Fig. 3 The influence of spontaneous variations in speed-accuracy tradeoff on metacognitive accuracy.** We investigated four different datasets (equaling a total $N = 430$) from experiments in which participants did not receive any instructions concerning speed or accuracy. Results from a hierarchical mixed effects model show that across the four datasets, there is a positive relation between M-ratio and v-ratio, b = 0.31, $p < 0.001$ (**A**), a negative relation between M-ratio and decision boundary, b = −0.200, $p = 0.015$ (**B**), and no relation between v-ratio and decision boundary, b = 0.11, $p = 0.233$ (**C**) Note: the regression line shows the estimate from the hierarchical model fit, transparent bands show the 95% confidence interval. Each dot reflects one participant, with the color depending on the dataset. Source data are provided as a Source Data file.

independent of response caution. Again, we tested whether these associations depend on confidence response mode. The full hierarchical mixed model in which we predicted v-ratio by M-ratio, decision boundary, and response mode and all interactions between these variables could not be estimated because the predictors were too strongly correlated (*Variance Inflation Factors* > 35). This was the case for M-ratio and its interaction with decision boundary, so we estimated the following reduced model:

$$\text{v-ratio} \sim \text{M-ratio} + \text{decision bound} \ast \text{confidence response mode} + (1|\text{experiment}) \quad (2)$$

As expected, the analyses showed a strong positive relation between M-ratio and v-ratio, b = 0.31, $t(424) = 5.548$, $p < 0.001$ (see Fig. 3A). Importantly, there was no main effect of decision boundary, $p = 0.233$, no main effect of confidence response mode, $p = 0.212$, nor an interaction between boundary and confidence response mode, $p = 0.166$. Note that the absence of a relation between v-ratio and decision boundary did not depend on the presence of M-ratio in the model: when dropping the M-ratio term from the model, results were virtually unchanged, all $ps > 0.350$. Finally, to show that the relation between v-ratio and M-ratio does not depend on response mode, we further fitted the following model:

$$\text{v-ratio} \sim \text{M-ratio} \ast \text{confidence response mode} + (1|\text{experiment}) \quad (3)$$

As expected, in this model we again found the relation between M-ratio and v-ratio, $p < 0.001$, but no main effect of confidence response mode, $p = 0.518$, nor an interaction effect, $p = 0.186$ (see Fig. 3C). Jointly, these two analyses show that v-ratio is an appropriate measure of metacognition because it is related to M-ratio but unrelated to the decision boundary.

For completeness, separate results of each experiment are described below and shown in Fig. 4, together with a visualization of model fit.

In Experiment 2 A (see Fig. 4A), we observed a positive but non-significant between-participants correlation between M-ratio and v-ratio, $r(61) = 0.21$, $p = 0.090$, a negative correlation between M-ratio and decision boundary, $r(61) = -0.41$, $p < 0.001$, but no relation between decision boundary and v-ratio, $r(61) = -0.03$, $p = 0.797$. The data showed a significant correlation between median choice RTs and median confidence RTs, $r(61) = 0.516$, $p < 0.001$, and between estimated decision boundaries and median confidence RTs, $r(61) = 0.440$, $p < 0.001$.

In Experiment 2B (see Fig. 4B), we observed a positive between-participants correlation between M-ratio and v-ratio, $r(94) = 0.63$, $p < 0.001$, no relation between M-ratio and decision boundary, $r(94) = 0.11$, $p = 0.282$, and a positive relation between decision boundary and v-ratio, $r(94) = 0.22$, $p = 0.029$. Finally, the data showed a significant correlation between median choice RTs and median confidence RTs, $r(94) = 0.490$, $p < 0.001$, and between estimated decision boundaries and median confidence RTs, $r(94) = 0.415$, $p < 0.001$.

In Experiment 2 C (see Fig. 4C), we observed a positive between-participants correlation between M-ratio and v-ratio, $r(202) = 0.277$, $p < 0.001$, a negative correlation between M-ratio and decision boundary, $r(202) = -0.16$, $p = 0.024$, and no relation between decision boundary and v-ratio, $r(202) = 0.109$, $p = 0.120$. Finally, the data showed a significant correlation between median choice RTs and median confidence RTs, $r(202) = 0.277$, $p < 0.001$, and between estimated decision boundaries and median confidence RTs, $r(202) = 0.250$, $p < 0.001$.

In Experiment 2D (see Fig. 4D), we observed no between-participants correlation between M-ratio and v-ratio, $r(65) = -0.019$, $p = 0.878$, a negative correlation between M-ratio and decision boundary, $r(65) = -0.30$, $p = 0.015$, but no relation between decision boundary and v-ratio, $r(65) = -0.18$, $p = 0.139$. Finally, the data showed no significant relation between median choice RTs and median confidence RTs, $r(65) = 0.228$, $p = 0.064$, and between estimated decision boundaries and median confidence RTs, $r(65) = 0.144$, $p = 0.244$.

**Relating decision boundary to *d'* and meta-*d'*.** Given that M-ratio reflects the ratio between *d'* and meta-*d'* it is instructive to further unravel the relation of both these measures with the decision boundary. Interestingly, whereas we observed a clear negative relation between M-ratio and decision boundary in both the model simulations and data, the findings concerning *d'* and meta-*d'* are much less straightforward. In the simulations, we observed a significant positive correlation between *d'* and decision boundary, $r(98) = 0.683$, $p < 0.001$, but not between meta-*d'* and decision boundary, $r(98) = -0.141$, $p = 0.160$. In Experiment 1, although M-ratio was modulated by instruction condition, this was not the case for *d'* ($M_{speed} = 1.47$ vs $M_{accuracy} = 1.66$), $p = 0.083$, or meta-*d'* ($M_{speed} = 1.05$ vs $M_{accuracy} = 0.84$), $p = 0.083$. Across the four datasets of Experiment 2, we observed no significant relation between *d'* and decision boundary, $b = -0.045$, $p = 0.444$, but we did observe a clear negative effect

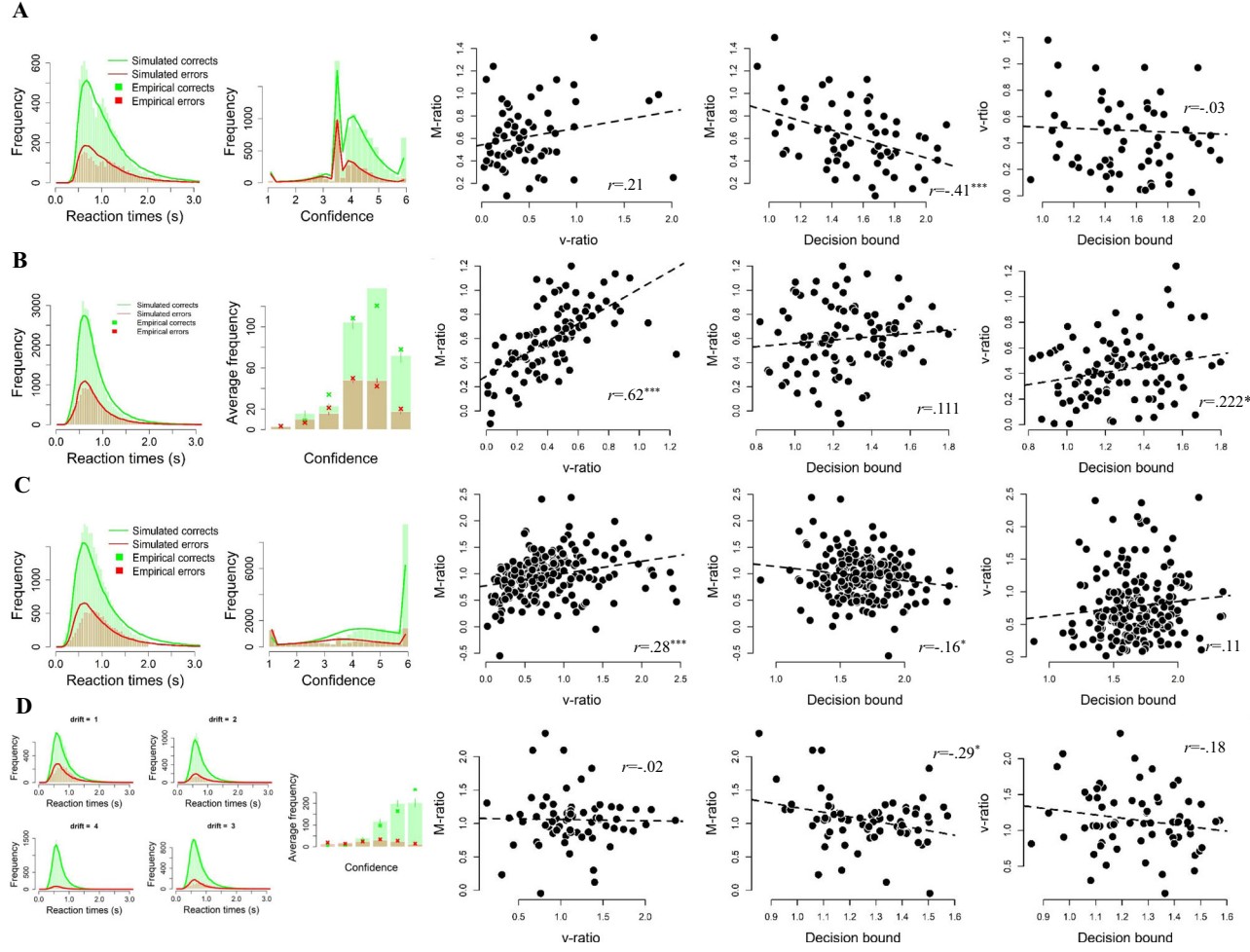

**Fig. 4 The influence of spontaneous variations in speed-accuracy tradeoff on metacognitive accuracy, separated by experiment.** Results are shown seperately for Experiment 2 A (panel A; N = 63), 2B (panel B; N = 96), 2 C (panel C; N = 204) and 2D (panel D; N = 67). Same convention as in Fig. 2; error bars reflect SEM. Source data areprovided as a Source Data file.

between meta-$d$' and estimated decision boundaries, $b = -0.358$, $t(427) = -3.527$, $p < 0.001$.

## Discussion

Researchers across several fields show an increasing interest in the question how observers can evaluate their task performance via confidence judgments. Crucial to study metacognition is a method to objectively quantify the extent to which participants are able to detect their own mistakes, regardless of decision strategy. We here report that a commonly used *static* measure of metacognitive accuracy (M-ratio) depends on the decision boundary – reflecting response caution – that is set for decision making. This was the case in simulation results, in two experiments explicitly manipulating the tradeoff between speed and accuracy, and across four datasets in which participants received no specific instructions concerning speed or accuracy. We propose an alternative, *dynamic*, measure of metacognitive accuracy (v-ratio) that does not depend on decision boundary.

**Caution is warranted with static measures of metacognition**. The most important consequence of the current findings is that researchers should be cautious when interpreting static measures of metacognitive accuracy, such as M-ratio. Although the findings reported in Experiments 2A-D are correlational and should thus

be interpreted with caution (e.g., it could be that participants with good metacognition deem it appropriate to impose low decision boundaries), the reported simulations and the within-participant experimental manipulation of Experiment 1 are indicative of a fundamental issue with M-ratio. Moreover, differences in confidence between the two instruction conditions in Experiment 1 were rather subtle, suggesting that even minor influences in decision confidence are sufficient to induce differences in M-ratio. As the name indicates, M-ratio reflects the ratio between meta-$d$' (second-order performance) and $d$' (first-order performance). Interestingly, whereas both the simulations and the experiments showed associations between M-ratio and decision boundary, the story was more complicated when instead directly relating meta-$d$' and $d$' with variations in decision boundary. Whereas in the simulations the relation between M-ratio and decision boundary was largely driven by $d$' but not so much by meta-$d$', these results were less clear in Experiment 1, whereas in Experiments 2A-D the effect appeared to be driven by meta-$d$'. Notably, our claim that signal-detection theoretic measures of performance are confounded by response caution applies to both second-order performance measures (e.g., meta-$d$') as well as first-order performance measures (e.g. $d$'). Everything else being equal, lower decision boundaries will lead to lower values of $d$', because choices will be made with less accumulated evidence. This knowledge, however, does not make the use of signal-detection

theoretic measures obsolete; indeed, its usefulness depends on the research question, experimental design, and other contextual factors. Likewise, the choice for M-ratio versus v-ratio as a measure of metacognitive accuracy might depend on similar considerations. Finally, it should be noted that although the data of Experiment 1 (and the replication reported in the Supplementary Information) showed that instructions to focus on speed vs accuracy influenced the fitted decision boundaries while leaving drift rates unaffected, this theoretically predicted pattern has not always been observed in previous work[35,36]. This is important, because if instructions to focus on speed would reduce the drift rate this will also have an influence on v-ratio because this measure reflects the ratio between drift rate and post-decisional drift rate. Therefore, it is important for future studies relying on the v-ratio framework to carefully consider the extent to which changes in metacognition between conditions or between participants are indeed driven by differences in post-decision drift rate, and not by non-selective changes in the drift rate.

In the following, we will discuss several examples where our finding might have important implications. In the last decade there has been quite some work investigating to what extent the metacognitive evaluation of choices is a domain-general process or not. These studies often require participants to perform different kinds of tasks, and then examine correlations in choice accuracy and in metacognitive accuracy between these tasks[3,17–20,37]. For example, Mazancieux and colleagues[17] asked participants to perform an episodic memory task, a semantic memory task, a visual perception task and a working memory task. In each task, participants rated their level of confidence after a decision. The results showed that whereas correlations between choice accuracy on these different tasks were limited, there was substantial covariance in metacognitive accuracy across these domains. Because in this study participants received no time limit to respond, it remains unclear whether this finding can be interpreted as evidence for a domain-general metacognitive monitor, or instead a domain-general response caution which caused these measures to correlate. Another popular area of investigation has been to unravel the neural signatures supporting metacognitive accuracy[19,20,38–40]. For example, McCurdy et al. observed that both visual and memory metacognitive accuracy correlated with precuneus volume, potentially pointing towards a role of precuneus in both types of metacognition. It remains unclear, however, to what extent differences in response caution might be responsible for this association. Although differences in response caution are usually found to be related to pre-SMA and anterior cingulate[24,25], there is some suggestive evidence linking precuneus to response caution[41]. Therefore, it is important that future studies on neural correlates of metacognition rule out the possibility that their findings are caused by response caution. Finally, our study may have consequences for investigations into differences in metacognitive accuracy between specific groups. For example, Folke and colleagues[23] reported that M-ratio was reduced in a group of bilinguals compared to a matched group of monolinguals. Interestingly, they also observed that on average bilinguals had shorter reaction times than monolinguals, but this effect was unrelated to the group difference in M-ratio. Because these authors did not formally model their data using evidence accumulation models, however, it remains unclear whether this RT difference results from a difference in boundary, and if so to what extent this explains the difference in M-ratio between both groups that was observed. In a similar vein, individual differences in M-ratio have been linked to psychiatric symptom dimensions, and more specifically to a symptom dimension related to depression and anxiety[5]. At the same time, it is also known that individual differences in response caution are related to a personality trait known as *need for closure*[42]. Given that need

for closure is, in turn, related to anxiety and depression[43], it remains a possibility that M-ratio is only indirectly related to these psychiatric symptoms via response caution.

**The potential of dynamic measures of metacognition.** In order to control for potential influences of response caution on measures of metacognitive accuracy, one approach could be to estimate the decision boundary and examine whether the relation between metacognitive accuracy and the variable of interest remains when controlling for decision boundary (e.g., using mediation analysis). However, a more direct approach would be to estimate metacognitive accuracy in a dynamic framework, thus directly taking into account differences in response caution. For example, building on the drift diffusion model it has been proposed that confidence reflects the level of integrated evidence following post-decisional evidence accumulation[27,32,34]. In the current work, we proposed v-ratio (reflecting the ratio between post-decision drift rate and drift rate) as such a dynamic measure of metacognitive accuracy (following the observation that post-decision drift rate indexes how accurate confidence judgments are[27,28]). In both simulations and empirical data, we observed a positive relation between v-ratio and M-ratio, suggesting they capture shared variance. Critically, v-ratio was not correlated with decision boundary, suggesting it is not affected by differences in response caution. Thus, our dynamic measure of metacognition holds promise as an approach to quantify metacognitive accuracy. An important caveat is that in order to measure v-ratio as accurately as possible, precise measurements of confidence reaction times are needed. This is an important concern, because in many metacognition experiments confidence is queried using approaches that do not provide precise measurements of confidence RTs. In fact, in several of the experiments reported in the current manuscript confidence was queried using a mouse or by moving a cursor along the scale using the keyboard arrows. Although the results of Experiment 1 and Experiments 2A-D did not depend on the mode of confidence responses, we strongly advise researchers interested in deploying v-ratio to collect data using a design that measures the timing of both choices and confidence in a very precise manner.

In our approach we allowed the drift rate and the post-decision drift rate to dissociate. This proposal is in line with the view of metacognition as a second-order process whereby dissociations between confidence and choice accuracy might arise because of noise or bias at each level[44–46]. However, when formulating post-decision drift rate as a continuation of evidence accumulation, it remains underspecified which evidence the post-decision accumulation process is exactly based on. It has been suggested that participants can accumulate evidence that was still in the processing pipeline (e.g. in a sensory buffer) even after a choice was made[34,47]. However, it is not very likely that this is the only explanation, particularly in tasks without much speed stress. One other likely possibility, is that during the post-decision process, participants resample the stimulus from short-term memory[48]. Because memory is subject to decay, dissociations between the post-decision drift rate and the drift rate can arise. Other sources of discrepancy might be contradictory information quickly dissipating from memory[49] which should decrease metacognitive accuracy, or better assessment of encoding strength with more time[50] which should increase metacognitive accuracy.

One important caution is that in our proposed formalization the computation of decision confidence (and thus metacognitive accuracy) arises by means of post-decisional evidence accumulation. Put simply, an observer with post-decision drift rate of one will be good at telling apart corrects from errors whereas an observer with post-decision drift rate of zero will be at chance

level. One way to evaluate the validity of post-decisional drift rate for this purpose, is to simulate data with various levels of post-decision drift rate, and then evaluate whether the area under the Type-II ROC is different from chance level. Type-II ROC analysis is a bias free measure that quantifies how well confidence tracks accuracy[16]. As discussed earlier, we indeed observed chance level area under type-II ROC whenever post-decision drift rate equals zero. Importantly, the choice to model decision confidence as a function of post-decisional evidence was directly informed by this finding. An alternative approach in the literature within the context of evidence accumulation models has been to quantify confidence as the probability of being correct given time, evidence and the response made[33,51–54]. Although this notion of decision confidence has been very successful in explaining empirical patterns seen in the literature, one drawback of this approach is that it does not predict chance-level type-II ROC performance with post-decision drift rate equal to zero, in the case of multiple drift rates. The reason that the probabilistic confidence model can still dissociate corrects from errors in this situation, is because it infers probability correct based on decision times (and both probability correct and decision times co-vary with drift rates). We therefore decided to quantify decision confidence as a function of post-decisional evidence, which allows to characterize v-ratio as a complete an unbiased measure of metacognitive accuracy.

Finally, we note that in the modeling efforts reported here, the duration of post-decisional evidence accumulation was decided based on the full distribution of empirically observed confidence RTs. Most previous modeling efforts have likewise assumed that post-decision processing terminates once confidence is externally queried[28], and only a few studies have explicitly examined different stopping rules for post-decision processing[32,33]. Given that we still lack a clear mechanistic understanding of how post-decisional processing is terminated, we here decided for this implementation which is agnostic regarding the underlying stopping criterion for confidence judgments, but nevertheless takes the full distribution of confidence RTs into account during fitting. By further unravelling the computational mechanisms underlying post-decisional accumulation termination, substantial progress can still be made by including these mechanisms in future modeling efforts.

To sum up, we provided evidence from simulations and empirical data that a common static measure of metacognition, M-ratio, is confounded with response caution. We proposed an alternative measure of metacognition based on a dynamic framework, v-ratio, which is insensitive to variations in caution, and may thus be suitable to study how metacognitive accuracy varies across participants and conditions.

## Methods
### Computational model
*Simulations.* Data were simulated for 100 observers with 1000 trials each. For each simulated observer, we randomly selected a value for the drift rate (uniform distribution between 1 and 3), for the decision boundary (uniform distribution between .5 and 4), for the non-decision time (uniform distribution between .2 and .6) and for the v-ratio (uniform distribution between 0 and 1.25; see below for details). When v-ratio is above 1, this implies that observers use more information to evaluate their choices compared to the actual choice[55]. To estimate meta-*d'*, data is needed for both of the possible stimuli (i.e., to estimate bias); therefore, for half of the trials we multiplied the drift rate by −1. During model fitting, we used the full distribution of empirically observed confidence RTs to inform the duration of post-decision processing time (see below). Given that the empirical data showed moderate correlations between average choice RTs and average confidence RTs (i.e., .277 and .516 for Experiment 2 and 3), for the model simulations we generated a distribution of post-decision processing times, the mean of which was moderately correlated with the mean of choice RTs. To achieve this, for each simulated observer we selected a value for boundary and drift rate from a normal distribution (sigma = 1) centered around the true boundary and true drift for that simulated observer, respectively. Then, we simulated a confidence RT distribution using these

two values (*ter* was set to 0 to account for the fact that confidence RTs are usually faster than choice RTs). During the actual simulations, post-decision processing times were sampled from this confidence RT distribution. This procedure induced a moderate correlation between average choice RTs and average confidence RTs, $r(98) = 0.537$, $p < 0.001$, and between average confidence RTs and decision boundary, $r(98) = 0.524$, $p < 0.001$. Finally, we fixed the values for starting point ($z = 0.5$) and within-trial noise ($\sigma = 1$). Note that the simulation results were very robust, as the same pattern of findings was obtained when increasing or decreasing the noise for generating dummy data, when restricting post-decision processing time to a fixed value, when only using a single drift rate for all simulations, and when simulating more observers (500), more trials per observer (5000), or both.

*Fitting procedure.* We coded an extension of the drift diffusion model (DDM) that simultaneously fitted choices, reaction times and decision confidence. The standard DDM is a popular variant of sequential sampling models of two-choice tasks. We used a random walk approximation, implemented in the rcpp R package to increase speed[56], in which we assumed that noisy sensory evidence started at $z*a$; 0 and a are the lower and upper boundaries, respectively, and z quantifies bias in the starting point ($z = 0.5$ means no bias). At each time interval $\tau$ a displacement $\Delta$ in the integrated evidence occurred according to the formula shown in Eq. (4):

$$\triangle = \nu * \tau + \sigma * \sqrt{\tau} * \mathcal{N}(0,1) \tag{4}$$

Evidence accumulation strength is controlled by $\nu$, representing the drift rate, and within-trial variability, σ, was fixed to 1. Note that it is common practice in DDM fitting to fix within-trial variability to 1, although this assumption of constant within-trial noise is often not made explicitly. The reason for fixing this parameter is that changes in σ cannot be dissociated from changes in a. Given that our hypothesis specifically concerns the latter, we decided to fix σ. The random walk process continued until the accumulated evidence crossed either 0 or a. After boundary crossing, the evidence continued to accumulate for a duration determined by the empirically observed confidence RT distribution (i.e., the difference in time between initial choice and confidence judgment). Specifically, the post-decision accumulation time of each simulated trial was set to be equal to the duration of a randomly selected trial from the confidence RT distribution of that participant. Note that this random selection was done without replacement, ensuring that the simulated confidence RT distribution exactly matched the empirically observed confidence RT distribution. Because the number of simulated trials always exceeded the number of empirical trials, sampling from the empirical confidence RT distribution restarted after all values were selected. Note that during the post-decisional accumulation period the integrated evidence was not limited between the boundaries (i.e. 0 was not a hard boundary) and could land anywhere between $-\infty$ and $+\infty$. Importantly, consistent with the signal detection theoretical notion that primary and secondary evidence can dissociate, we allowed for dissociations between the drift rate governing the choice and the post-decision drift rate. For compatibility with the M-ratio framework, we quantified metacognitive accuracy as the ratio between post-decision drift rate and drift rate, as shown in Eq. (5):

$$\text{v-ratio} = \frac{\nu\, post}{\nu} \tag{5}$$

As a consequence, when v-ratio = 1, this implies that post-decision drift and drift are the same. When v-ratio = 0.5, the magnitude of the post-decision drift rate is half the magnitude of the drift rate. To calculate decision confidence, we assumed a direct mapping between post-decisional evidence and decision confidence. To take into account idiosyncratic mappings between evidence and the confidence scale used in the experiment, we added two extra free parameters that controlled the mean (M) and the width (SD) of confidence estimates, as shown in Eq. (6):

$$confidence = \frac{e_{t+s,X} + M}{SD} \tag{6}$$

In the experiments measuring confidence using a continuous scale, there was an over-representation of confidence values at the boundaries (i.e. 1 and 6) and in the middle of the scale (50 in Experiment R1 reported in the Supplementary Information, 3.5 in Experiment 2 A). Most likely, this resulted from the use of verbal labels placed at exactly these values. To account for peaks in the center of the scale, we assumed that confidence ratings around the center were pulled towards the center value. Specifically, we relabeled P% of trials around the midpoint as the midpoint (e.g., in Experiment R1, $P = 10\%$ implies that 10% of the data closest to 50 were (re)labeled as 50). Note that P was not a free parameter, but instead its value was taken to be the participant-specific proportion based on the empirical data. In the experiments measuring confidence using a six point scale, after applying Eq. 6, predicted confidence was divided into six categories by means of rounding. To account for frequency peaks at the endpoints of the scale, we relabeled predicted confidence values that exceeded the endpoints of the scale as the corresponding endpoint (e.g., in Experiment 1 a predicted confidence value of 7 was relabeled as 6), which naturally accounted for the frequency peaks at the endpoints. Note that the main conclusions reported in this manuscript concerning the relation between M-ratio, decision boundary and post-decision drift rate, remain the same in a model without P, and also in a reduced model without P, M

and *SD*. Because these reduced models did not capture confidence distributions very well though, we here report only the findings of the full model.

To estimate these 6 parameters (*v, a, Ter, v-ratio, M,* and *SD*) based on choices, reaction times and decision confidence, we implemented quantile optimization. Specifically, we computed the proportion of trials in quantiles .1, .3, .5, .7, and .9, for both reaction times and confidence; separately for corrects and errors (maintaining the probability mass of corrects and errors, respectively). We then used differential evolution optimization, as implemented in the DEoptim R package[57], to estimate these 6 parameters by minimizing the chi square error function shown in Eq. (7):

$$x^2 = \sum \frac{(oRT_i - pRT_i)^2}{pRT_i} + \sum \frac{(oCJ_i - pCJ_i)^2}{pCJ_i} \qquad (7)$$

with $oRT_i$ and $pRT_i$ corresponding to the proportion of observed/predicted responses in quantile *i*, separately calculated for corrects and errors both reaction times, and $oCJ_i$ and $pCJ_i$ reflecting their counterparts for confidence judgments. Note that in the experiments in which confidence was queried using a 6-point scale, we computed the proportion of datapoints per level of confidence, rather than per quantile. In Experiment 2D, there were four levels of difficulty (by crossing 2 levels of mean with 2 levels of variance), and so four drift rates were estimated, one per condition. In all experiments, we set $\tau$ to 0.001. Model fitting was done separately for each participant. For all experiments, choice RTs faster than 100 ms and confidence RTs slower than 5 s were excluded for fitting purposes. M-ratio, reflecting the ratio between meta-*d*' and *d*', was calculated using the method described in Maniscalco and Lau[15] which adjusts for unequal variances. Note that the findings regarding M-ratio remain unchanged without this correction, except for Experiment 1 where the difference in M-ratio between the two instruction conditions becomes $p = 0.053$.

*Parameter recovery*. To assure that our model was able to recover the parameters, we here report parameter recovery. In order to assess parameter recovery with a sensible set of parameter combinations, we used the fitted parameters of Experiment 1, simulated data from these parameters with a varying number of trials, and then tested whether our model could recover these initial parameters. As a sanity check, we first simulated a large number of trials (25,000 trials per participant), which as expected provided excellent recovery for all six parameters, $rs > 0.97$. We then repeated this process with only 200 trials per participants, which was the trial count in Experiment 2 A (note that the other experiments had higher trial counts). Recovery for v-ratio was still very good, $r = 0.85$, whereas it remained excellent for all other parameters, $rs > 0.98$.

**Experiment 1**. All research reported here complies with the relevant ethical regulations, ethical approval has been obtained from the local ethics committee of the KU Leuven. Participants in Experiment 1, Experiment 1 S, and Experiment 2B all provided informed consent before their participation. Ethical approval for these studies was granted by the Social and Societal Ethics Committee of the KU Leuven. The other experiments were reanalyses of previously published data.

*Participants*. Forty-three healthy participants (16 males) took part in Experiment 1 in return for course credit (mean age = 19.2, between 18 and 22). All reported normal or corrected-to-normal vision. One participant was excluded because they required more than 10 practice blocks in one of the training blocks (see below) and eight participants were excluded because their choice accuracy was not different from chance level performance in at least one of both instruction conditions, as assessed using a chi square test. Finally, two participants were excluded because they use the same confidence button in more than 95% of trials. The final sample thus comprised thirty-two participants. All participants provided their informed consent and all procedures adhered to the general ethical protocol of the Social and Societal Ethics Committee of the KU Leuven.

*Stimuli and apparatus*. The data for Experiment 1 were collected in an online study, due to COVID-19, using jsPysch library[58]. As a consequence, caution is warranted when interpreting between-participant differences given that we had no control over hardware specifications. Given that Experiment 1 concerns a within-participant experimental manipulation, however, the risk that this is problematic for our data is small. Both choices and confidence judgments were provided with the keyboard. Stimuli in Experiment 1 consisted of 50 randomly moving white dots (radius: 2 pixels) drawn in a circular aperture on a black background centered on the fixation point. Dots disappeared and reappeared every 5 frames. The speed of dot movement (number of pixel lengths the dot will move in each frame) was a function of the screen resolution (screen width in pixels / 650).

*Task procedure*. Each trial started with the presentation of a fixation cross for 1000 ms. Above and below this fixation cross specific instructions were provided concerning the required strategy. In accuracy blocks the instruction was to respond as accurately as possible; in speed blocks the instruction was to respond as fast as possible. The order of this block-wise manipulation was counterbalanced across participants. Next, randomly moving dots were shown on the screen until a response was made or the response deadline was reached (max 5000 ms). On each

trial, 20% of the dots coherently moved towards the left or the right side of the screen, with an equal number of leftward and rightward movement trials in each block. Participants were instructed to decide whether the majority of dots was moving towards the left or the right side of the screen, by pressing "C" or "N", respectively, with their left or right thumb, respectively. After their response, a 6-point confidence scale was presented on the screen. The numerical keys '1', '2', '3', '8', '9', and '0' on top of the keyboard mapped onto 'certainly correct', 'probably correct', 'guess correct', 'guess wrong', 'probably wrong', and 'certainly wrong', respectively (mapping counterbalanced between participants). The labels of this confidence scale, ranging from 'certainly wrong' to 'certainly correct', might appear unusual to readers familiar with confidence scales where the lowest part of the scale corresponds to 'guessing'. Although the lower part of the scale (e.g. 'certainly wrong') is used very infrequently, in previous research we have documented that such confidence judgments do reflect genuine experiences and are associated with unique compensatory behavior[59]. Although a scale using a smaller range of confidence judgments would make it appear as if the ratings are not that compressed (and more spread across the scale), this comes with the risk that trials on which participants detect themselves making an error cannot be judged with the appropriate level of confidence.

The main part of Experiment 1 consisted of 10 blocks of 60 trials, half of which were from the accuracy instruction condition and half from the speed instruction condition. The experiment started with 24 practice trials during which participants only discriminated random dot motion at 50% coherence, no confidence judgments were asked. This block was repeated until participants achieved 85% accuracy (mean = 1.97 blocks). Next, participants completed again 24 practice trials with the only difference that now the coherence was decreased to 20% (mean = 1.09 blocks). When participants achieved 60% accuracy, they performed a final training block of 24 trials during which they practiced both dot discrimination and indicated their level of confidence (mean = 1.26 blocks).

**Experiment 2A**. Full experimental details are described in Drescher et al.[60]. Ethical approval was obtained in the original study, and participants provided written informed consent before participation. Participants were seated in individual cubicles in front of a 15-in CRT monitor with a vertical refresh rate of 85 Hz. On each trial participants were presented with two white circles (5.1° diameter) on a black background, horizontally next to each other with a distance of 17.8° between the midpoints. Fixation crosses were shown for 1 s in each circle, followed by dots clouds in each circle for 700 ms. The dots had a diameter of 0.4°. Dot positions in the boxes, as well as the position of the box containing more dots were randomly selected on each trial. The difference in number of dots between both boxes (indexing task difficulty) was adapted online using an unequal step size staircase procedure[20]. Participants indicated which circle contained more dots by pressing "S" or "L" on a keyboard. Then, the question "correct or false?" appeared on the screen, with a continuous confidence rating bar, with the labels "Sure false", "No idea", and "Sure correct". Participants moved a cursor with the same keys as before, which they could do by holding down one of both keys which moved the cursor along the scale, and confirmed their confidence judgment with the enter key. No time limit was imposed for both primary choice and confidence rating. Participants received several practice trials (10 without confidence rating, 14 with confidence rating), before they completed eight experimental blocks of 25 trials.

**Experiment 2B**. The data for Experiment 2B were collected in an online study, due to COVID-19. Ethical approval was obtained from the Social and Societal Ethics Committee at KU Leuven, and participants provided informed consent before participation. As a consequence, caution is warranted when interpreting between-participant differences given that we had no control over hardware specifications. Ninety-nine participants (mean age = 18.5, range 18–21; 10 male) took part in return for course credit. Three participants were excluded because they used the same level of confidence for more than 90% of their choices, leaving a total of 96 participants. The experiment was similar to Experiment 2 A, except for the following: dots were presented for 300 ms inside two squares. Choices were indicated by pressing "V" or "N" with the thumbs of both hands, and confidence was indicated in the same way as in Experiment 1. Participants performed 10 blocks of 50 trials, after first completing 26 practice trials.

**Experiment 2C**. Data from this experiment were taken from the confidence database[61], a collection of openly available studies on decision confidence. In this experiment, Prieto, Reyes and Silva[62], used the same task as described in Fleming and colleagues[2]. Ethical approval was obtained in the original study, and participants provided written informed consent before participation. Each participant (N = 204, all female, aged 18–35) completed 50 practice trials, followed by 5 blocks of 200 trials. On each trial participants were presented with an array of eight Gabor patches for 200 ms, a blank for 300 ms and another array of 6 Gabor patches for 200 ms. Participants had to decide whether the first or the second temporal interval contained a patch with a higher contrast. The contrast of the pop-out Gabor was continuously adapted using an online staircase procedure to maintain 71% accuracy. Choices and confidence reports were collected in an identical manner as in Experiment 2. The only difference was that there was no verbal description around the middle point of the confidence scale.

**Experiment 2D**. The data of Experiment 2D comprise four different datasets using exactly the same design, coming from Boldt et al.[63], Boldt et al.[64] and Desender et al. Experiment 3B[59], equaling a total $N = 67$. Full experimental details are described in Boldt et al.[63]. Ethical approval was obtained in the original studies, and participants provided written informed consent before participation. After a fixation point for 200 ms, the stimulus was flashed for 200 ms, followed again by the fixation point. The experiment manipulated the average color (two levels: high vs low) of the eight elements and the variance across the colors (two levels: high vs low). Participants were instructed to decide whether the average color of the eight elements was blue or red, using the same response lay-out as in Experiment 1. When participants did not respond within 1500 ms, the trial terminated and the message 'too slow, press any key to continue' was shown. When participants responded in time, a fixation point was shown for 200 ms, followed by a confidence prompt using the same layout as Experiment 1. The inter-trial interval lasted 1000 ms. Each block started with 12 practice trials with auditory performance feedback in which the confidence judgment was omitted. The experiment started with one practice block (60 trials) without confidence judgments but with auditory performance feedback and one practice block (60 trials) with confidence judgments but without feedback.

**Analysis**. Whenever applicable, statistical tests were always two-tailed

**Reporting summary**. Further information on research design is available in the Nature Research Reporting Summary linked to this article.

## Data availability

All raw data have been deposited online and can be freely accessed (github.com/kdesende/dynamic_influences_on_static_measures/), except for the data of Experiment 2 A which can be found elsewhere (https://github.com/l-drescher/raw_data_MW_MC_CC) and the data of Experiment 2 C which is part of the Confidence Database (https://osf.io/s46pr/)[61]. Note that Experiment 2 A is a reanalysis of Drescher et al.[60], and Experiment 2D is a reanalysis of Boldt et al.[63], Boldt et al.[64], and Desender et al. Experiment 3B[59]. Source data are provided with this paper.

## Code availability

All analysis code have been deposited online and can be freely accessed (github.com/kdesende/dynamic_influences_on_static_measures/)[65].

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

## Acknowledgements
The authors like to thank Peter R Murphy, Bharath Chandra Talluri, and Annika Boldt for insightful discussions and Pierre Le Denmat, Robin Vloeberghs and Alan Voodla for comments on an earlier draft. This research was supported by an FWO [PEGASUS]² Marie Skłodowska-Curie fellowship (12T9717N, to K.D.) and a starting grant from the KU Leuven (STG/20/006, to K.D.). L.V. was supported by the Research Foundation - Flanders (FWO-Vlaanderen) (11H5619N).

## Author contributions
K.D., L.V. and T.V. designed the studies. K.D. and L.V. developed the model and collected the data. K.D. analyzed model simulations and data. K.D. wrote the manuscript. K.D., L.V. and T.V. discussed the results and commented on the manuscript.

## Competing interests
The authors declare no competing interests.
