## [Peer Review File · Nature Communications]

nature portfolio

Peer Review File

Dynamic influences on static measures of metacognitionEditorial Note: This manuscript has been previously reviewed at another journal that is not operating a transparent peer review scheme. This document only contains reviewer comments and rebuttal letters for versions considered at Nature Communications.

Reviewers' comments: Reviewer #1 (Remarks to the Author):

The authors have adequately addressed my comments.

Reviewer # 2 (Remarks to the Author):

Thank you for considering my original comments carefully. I have a couple of follow-up remarks based on some of the responses.

Original point 1: thank you for the analysis on the RT distributions; this looks quite convincing in terms of separating out premature errors from those resulting from low signal evidence. Having said this, however, I would argue that though SAT changes can be explained largely by boundary adjustments, there is still ample evidence in the literature pointing to accompanying drift rate changes. As such the larger the accuracy differences across conditions the more likely these drift rate effects would be to shine through. In the absence of such effects in your data, would you be able to provide instead some intuition/discussion on how this would impact the estimation of v -ratio under the two instructions? In other words, my original question 1-ii still stands.

Original point 2. Here I will disagree that because only a single noise level was used then "it should not come as a surprise that the distribution of confidence ratings are rather compressed". This will depend entirely on how that noise level was chosen. If the evidence is too high then one might observe what has been seen in this work (compressed ratings). If on the other hand it's chosen so that the task is more challenging then the spread of confidence ratings would be higher. The latter would be preferable since one can capture a fully graded range of confidence levels for otherwise nominally identical stimuli. In the absence of this, the analysis presented in the rebuttal is a reasonable compromise and the accompanying description quite useful in better appreciating how the method behaves under these conditions. However, the one sentence that was added to the text in response to this comment isn't providing a satisfactory account of the intuition provided in the response letter itself (which I think would be useful for the reader). Please consider expanding this further.

Reviewer # 3 (Remarks to the Author):

I very much appreciate the authors' extensive and responsive reply to my original review and commend them for their effort and thoroughness. However, in spite of these helpful clarifications and extra analyses, I still find several key aspects of the paper's argument to be lacking, primarily due to fundamental limitations in the data sets used in relation to the claims the authors want to make.

A really central limitation of the current manuscript remains the issue of measuring and modeling confidence RTs. The reason this is such a central issue is that the authors' modeling approach proposes to measure metacognitive accuracy as post-decision drift rate, but drift rate cannot be meaningfully fit to data in the absence of considering accumulation time; the two go hand in hand. As a consequence, the modeling and measurement of v -ratio is only as solid as the modeling and measurement of post-decision RT.

The authors take this into account in the modeling by setting post-decision accumulation time to a constant value. In fitting the model to a subject's data, the post-decision accumulation time is set so as to reflect the subject's median confidence RT. In the model simulations, post-decision reaction time is chosen so as to mimic the empirical correlations between choice RTs and confidence RTs. Thus the simulated data match the median of empirical confidence RT but without any corresponding trial-to-trial variability. While this modeling choice can be framed as a reasonable simplification, a really full drift diffusion model treatment would involve explicitly modeling confidence RT distributions in the same way that classic DDMs model distributions of choice RT. Such an approach would provide extra constraints on the fitting of v -ratio that could

affect fitting results and interpretation. The authors' simpler approach is reasonable as a first pass, but may be neglecting important facets of the data, which could in turn influence the fitted values of v_ratio and ultimately, interpretation of the modeling fits.

However, a far more severe limitation than the reasonable model simplifications noted above is the quality of the confidence RT data. In Experiments 1-3, subjects entered perceptual decisions with a single key press, but entered confidence rating on a continuous scale, either with a mouse click (Expt 1) or by using keys to navigate a cursor on the scale (Expts 2-3). In all cases, the mean and variance of the motor component required to enter the confidence rating was likely very large compared to both (1) the corresponding motor component of key pressing in entering the perceptual decision, and (2) the actual decision time for deciding on a confidence rating. This seems to be reflected in the magnitude of the confidence RTs in Expts 1-3, as many subjects exhibit conf RTs of 1 s or larger, which is 2-3 times larger than confidence RTs I have typically encountered in data where subjects enter confidence on a discrete (e.g. 2-point or 4-point) rating scale using key presses. (Though I do recognize that longer conf RTs could be in part due to the added decisional burden of entering confidence on a continuous scale rather than a discrete scale.) In Expts 2-3, since a cursor on the confidence scale was adjusted by key presses, conf RT was also likely confounded with confidence magnitude, with very low and very high confidence trials presumably taking longer for the cursor to finally arrive at the correct position on the scale, thus artificially inflating conf RT for more extreme confidence ratings.

In a nutshell, I am not convinced that the designs of these experiments allows for meaningful interpretation of the confidence RT data. This is not a shortcoming of analysis or interpretation but rather one of the suitability of these experimental designs and data, relative to the intended purposes of this paper. I raised this concern in my previous review but the authors' response to this particular point was not very elaborate or compelling. I will go to greater lengths here to explain why I think it's an issue of central importance for this work, and therefore can't be put off for future work to address.

A central part of the authors' argument is that M_ratio correlates with fitted decision bounds, whereas v_ratio doesn't. The implication is that in these data, M_ratio is confounded with a parameter related to perceptual decision making (decision bound) rather than metacognitive evaluation of accuracy per se, whereas v_ratio isn't. However, the fact that M_ratio correlates with decision bound doesn't necessarily imply that the correlation is spurious; it could be that in Expts 2-3, subjects who were better at the task (i.e. lower decision bound needed in order to achieve target level of task performance) really were also better at metacognitive evaluation as well. Even in the within-subject design of Expt 1, it could be that the different mental set adopted in the "speed" condition induced changes that manifested partially as real differences in metacognitive ability. The correlations alone are insufficient to determine which interpretation is more likely to be correct. Thus, the entire rationale for interpreting the M_ratio correlations as spurious comes down to contrasting these with the non-significant correlations of v_ratio with decision bound. However, these v_ratio results themselves depend on a simplified modeling approach that only considers median confidence RT rather than full confidence RT distributions, and more crucially, depend on conf RT data that are likely heavily influenced by a large motor latency/noise component that may (1) obscure the decisional component of conf RT, (2) obscure the relationship between conf RT and choice RT. This issue with the quality of the conf RT data then casts doubt on the v_ratio fits, which in turn casts doubt on the interpretation of the M_ratio results, and ultimately casts doubt on the main argument of the paper.

The authors propose to address these points about confidence RT in future modeling work, but in the absence of addressing them satisfactorily for the *present* work, it cannot stand as a really strongly held together scientific argument and instead comes off as an intriguing idea that is still in need of strong empirical support and (ideally) further sophistication in the modeling. Characterizing drift rate well requires characterizing the corresponding RTs well, so measuring and modeling confidence RT is something the arguments of this paper can't afford not to get right if it's going to be a really strong and compelling work.

Other points

* Model simulation: RT correlations and M_{ratio}

I appreciate the authors' revision of the model simulations so as to roughly reproduce the correlation coefficients between choice RT and conf RT seen in Expts 1-3. However, this only partially addresses the issue, since correlation does not take scaling into account. (e.g. the correlation between X and Y is the same as the correlation between $10 \cdot X$ and $100 \cdot Y$.) The practical concern here is that even if the simulated choice RT and conf RT have similar correlation to what is seen in the empirical data, this doesn't ensure that the actual *magnitudes* of the conf RTs in relation to the choice RTs will reflect the empirical patterns. It appears as though the simulated choice and conf RTs do in fact have roughly similar relative magnitudes as those found in the data, but this was not highlighted in the relevant discussion and is not entailed by the similar correlation coefficient alone. For instance, had all the simulated conf RTs been multiplied by 100, that would still yield the same correlation coefficient between simulated choice RT and conf RT, even though it obviously would not be a good reflection of the data.

A more substantive point about the simulated data is that many simulated data points have implausibly high M_{ratio} values. M_{ratio} is typically observed to be close to 1, roughly in line with theoretical expectation. Values greater than 1 do occur empirically, but values above 2 are almost never seen except in outlier cases where accurate estimation becomes difficult (e.g. due to low N or very low values of d'). This is born out in Expt 1-3, where M_{ratio} tops out at about 1-2. Yet the simulations involve many unrealistic cases where M_{ratio} ranges from 2-4. It appears that inclusion of these extreme M_{ratio} values artificially influences the correlation with both simulated v_{ratio} and simulated decision bound (Fig 1C). The authors should choose simulation parameters that do not yield M_{ratio} values above 2, and ideally have the majority of cases closer to or below $M_{ratio} = 1$, more in line with the range of M_{ratio} values typically observed in real data that does not suffer from statistical estimation issues.

* rating confidence from $p(\text{correct})$

The authors have nicely shown that computing confidence from $p(\text{correct} | e, t+s, X)$ does not entail that area under the type 2 ROC > 0.5 when $v_{ratio}=0$, or when post-decision accumulation time $e=0$. However, they show that this is only the case when simulating data with a single drift rate, and does not hold when simulating with two drift rates.

I suspect that the reason for this pattern of results is that, in DDMs where drift rate is constant, the RT distributions for correct and incorrect responses are the same-- which fails to capture the common empirical pattern whereby correct responses have faster RTs. This RT pattern can be captured, however, by introducing trial-by-trial variability in drift rate (Ratcliff & Rouder, 1998). Thus, in the authors' simulations with a single (non-varying) drift rate, RT distributions for correct and incorrect responses are likely the same, entailing that estimation of $p(\text{correct})$ carries no useful information when $v_{ratio}=0$ or post-decision accumulation time $=0$ -- hence, type 2 AUC = 0.5. Whereas simulating two drift rates introduce some drift rate variability, and therefore differences in RT distributions for correct and incorrect trials that can be used to diagnose accuracy to some extent even when $v_{ratio}=0$ or post-decision accumulation time $=0$.

Thus, I think the point raised in my original review still stands-- there is a conceptual tension between (1) estimating confidence from $p(\text{correct})$ in the way the authors do, and (2) characterizing metacognitive accuracy using v_{ratio} , since on this formulation v_{ratio} does not contain all the relevant information entering into confidence ratings. The authors' demonstration that type 2 AUC = 0.5 when $v_{ratio}=0$ or post-decision accumulation time $=0$ thus only reflects the fact that their current modeling choices cannot account for differences in RT for correct and incorrect trials, which is a shortcoming. A fuller model implementation that could account for such RT differences might well exhibit type 2 AUC appreciably above-chance even when $v_{ratio}=0$ or post-decision accumulation time $=0$. (Such a model would presumably have more drift variance than the authors' two drift rate simulation, and therefore might have type 2 AUCs appreciably larger than the 0.511 value found for the two drift rate simulation.) This would then reintroduce the conceptual tension that v_{ratio} is not measuring everything there is to metacognitive accuracy after all, and at bottom the real work is being done by the unexplained $p(\text{correct})$ calculation.

The authors write, "Finally, we feel that, rather than it being our aim to dissociate between these two very similar models, the goal of the current work is to demonstrate that static models of metacognition are too simplistic, and that instead dynamic models should be used." I take the point that the $p(\text{correct})$ issue is not the most central issue to address, but it is one that still seems in need of addressing nonetheless. And it is not a matter of dissociating the $p(\text{correct})$ version of the model from the more purely Pleskac & Busemeyer-type version of the model, so much as it is a matter of choice and conceptual interpretation. If the authors want to characterize v_{ratio} as a *complete* measure of metacognitive accuracy, this does not seem to leave room for the $p(\text{correct})$ implementation of the model which allows for influences on metacognitive accuracy outside of v_{ratio} . Conversely, if the authors want to use the $p(\text{correct})$ version of the model, this does not seem to leave room for interpreting v_{ratio} as a complete measure of metacognitive accuracy. I think either choice is viable, but just want to point out that the authors *do* have a choice to make here and can't have their cake and eat it too.

Reviewer #1

The authors have adequately addressed my comments.

We would like to thank Reviewer 1 for their appreciation of our previous revision.

Reviewer #2 (note, this was Rev3 in the previous round, and vice versa)

Thank you for considering my original comments carefully. I have a couple of follow-up remarks based on some of the responses.

Original point 1: thank you for the analysis on the RT distributions; this looks quite convincing in terms of separating out premature errors from those resulting from low signal evidence. Having said this, however, I would argue that though SAT changes can be explained largely by boundary adjustments, there is still ample evidence in the literature pointing to accompanying drift rate changes. As such the larger the accuracy differences across conditions the more likely these drift rate effects would be to shine through. In the absence of such effects in your data, would you be able to provide instead some intuition/discussion on how this would impact the estimation of v-ratio under the two instructions? In other words, my original question 1-ii still stands.

We agree with the Reviewer that it is unfortunate that in Experiment 1 there was no significant difference in accuracy between both speed-accuracy tradeoff instruction conditions. Importantly, as suggested by the Editor we replicated this experiment, with the only difference that now confidence ratings were given with discrete button presses. In this new dataset ($N = 32$), we did observe significant differences for both reaction times, $p < .001$, and accuracy, $p = .035$, when comparing both instruction conditions. In line with the previously reported experiment, this difference was largely captured by changes in decision boundary, $p < .001$, whereas drift rate was again not significantly different, $p = .478$. Thus, we have provided clear evidence that, at least in our experiments, participants selectively change their decision boundary when instructed, leaving drift rate unaffected. Nevertheless, we agree with the reviewer that such a pattern has not been consistently observed in the literature, and we now discuss the implications of this on p. 14:

„Moreover, it should be noted that although the data of Experiment 1 (and the replication reported in the Supplementary Materials) showed that instructions to focus on speed vs accuracy influenced the fitted decision boundaries while leaving drift rates unaffected, this theoretically predicted pattern has not always been observed in previous work 37,38. This is important, because if instructions to focus on speed would reduce the drift rate this will also have an influence on v-ratio because this measure reflects the ratio between drift rate and post-decisional drift rate. Therefore, it is important for future studies relying on the v-ratio framework to carefully consider the extent to which changes in metacognition between conditions or between participants are indeed driven by differences in post-decision drift rate, and not by non-selective changes in the drift rate“

Original point 2. Here I will disagree that because only a single noise level was used then “it should not come as a surprise that the distribution of confidence ratings are rather compressed”. This will depend entirely on how that noise level was chosen. If the evidence is too high then one might observe what has been seen in this work (compressed ratings). If on the other hand it’s chosen so that the task is more challenging then the spread of confidence ratings would be higher. The latter would be preferable since one can capture a fully graded range of confidence levels for otherwise nominally identical stimuli. In the absence of this, the analysis presented in the rebuttal is a reasonable compromise and the accompanying description quite useful in better appreciating how the method behaves under these conditions. However, the one sentence that was added to the text in response to this comment isn’t providing a satisfactory account of the intuition provided in the response letter itself (which I think would be useful for the reader). Please consider expanding this further.

As requested by the Reviewer, we have now extended the discussion about the compressions of confidence judgments in the manuscript itself. The relevant paragraph can be found on p. 22:

„The labels of this confidence scale, ranging from ‘certainly wrong’ to ‘certainly correct’, might appear unusual to readers familiar with confidence scales where the lowest part of the scale corresponds to ‘guessing’. Although the lower part of the scale (e.g. ‘certainly wrong’) is used very infrequently, in previous research we have documented that such confidence judgments do reflect genuine experiences and are associated with unique compensatory behavior⁵⁷. Although a scale using a smaller range of confidence judgments would make it appear as if the ratings are not that compressed (and more spread across the scale), this comes with the risk that trials on which participants detect themselves making an error cannot be judged with the appropriate level of confidence.“

Reviewer #3

I very much appreciate the authors' extensive and responsive reply to my original review and commend them for their effort and thoroughness. However, in spite of these helpful clarifications and extra analyses, I still find several key aspects of the paper's argument to be lacking, primarily due to fundamental limitations in the data sets used in relation to the claims the authors want to make.

A really central limitation of the current manuscript remains the issue of measuring and modeling confidence RTs. The reason this is such a central issue is that the authors' modeling approach proposes to measure metacognitive accuracy as post-decision drift rate, but drift rate cannot be meaningfully fit to data in the absence of considering accumulation time; the two go hand in hand. As a consequence, the modeling and measurement of v_ratio is only as solid as the modeling and measurement of post-decision RT.

The authors take this into account in the modeling by setting post-decision accumulation time to a constant value. In fitting the model to a subject's data, the post-decision accumulation time is set so as to reflect the subject's median confidence RT. In the model simulations, post-decision reaction time is chosen so as to mimic the empirical correlations between choice RTs and confidence RTs. Thus the simulated data match the median of empirical confidence RT but without any corresponding trial-to-trial variability. While this modeling choice can be framed as a reasonable simplification, a really full drift diffusion model treatment would involve explicitly modeling confidence RT distributions in the same way that classic DDMs model distributions of choice RT. Such an approach would provide extra constraints on the fitting of v_ratio that could affect fitting results and interpretation. The authors' simpler approach is reasonable as a first pass, but may be neglecting important facets of the data, which could in turn influence the fitted values of v_ratio and ultimately, interpretation of the modeling fits.

We agree with the Reviewer that our choice to set post-decision accumulation time to a fixed value was a reasonable simplification, but given the important consequences that our work will have for the field of metacognition, we appreciate the concern about this being potentially an oversimplification. Therefore, as requested by the Reviewer, we further improved our modelling efforts and now explicitly fitted our computational model to the *entire* confidence RT distributions (instead of a single summary metric). Importantly, this improvement to our model did not affect any of our conclusions, i.e. we still observed a consistent negative association between decision boundary and M-ratio, but not with v -ratio. This key finding was again observed in all three earlier reported experiments, in the newly collected dataset (see below), and in the updated model simulations. With this improved model, we are convinced that we have satisfied any remaining concerns about the quality of our computational model framework. We do not reiterate all these results here, but we invite the reader to inspect these in the revised manuscript.

However, a far more severe limitation than the reasonable model simplifications noted above is the quality of the confidence RT data. In Experiments 1-3, subjects entered perceptual decisions with a single key press, but entered confidence rating on a continuous scale, either with a mouse click (Expt 1) or by using keys to navigate a cursor on the scale (Expts 2-3). In all cases, the mean and variance of the motor component required to enter the confidence rating was likely very large compared to both (1) the corresponding motor component of key pressing in entering the perceptual decision, and (2) the actual decision time for deciding on a confidence rating. This seems to be reflected in the magnitude of the confidence RTs in Expts 1-3, as many subjects exhibit conf RTs of 1 s or larger, which is 2-3 times larger

than confidence RTs I have typically encountered in data where subjects enter confidence on a discrete (e.g. 2-point or 4-point) rating scale using key presses. (Though I do recognize that longer conf RTs could be in part due to the added decisional burden of entering confidence on a continuous scale rather than a discrete scale.) In Expts 2-3, since a cursor on the confidence scale was adjusted by key presses, conf RT was also likely confounded with confidence magnitude, with very low and very high confidence trials presumably taking longer for the cursor to finally arrive at the correct position on the scale, thus artificially inflating conf RT for more extreme confidence ratings.

In a nutshell, I am not convinced that the designs of these experiments allows for meaningful interpretation of the confidence RT data. This is not a shortcoming of analysis or interpretation but rather one of the suitability of these experimental designs and data, relative to the intended purposes of this paper. I raised this concern in my previous review but the authors' response to this particular point was not very elaborate or compelling. I will go to greater lengths here to explain why I think it's an issue of central importance for this work, and therefore can't be put off for future work to address.

*A central part of the authors' argument is that M_ratio correlates with fitted decision bounds, whereas v_ratio doesn't. The implication is that in these data, M_ratio is confounded with a parameter related to perceptual decision making (decision bound) rather than metacognitive evaluation of accuracy per se, whereas v_ratio isn't. However, the fact that M_ratio correlates with decision bound doesn't necessarily imply that the correlation is spurious; it could be that in Expts 2-3, subjects who were better at the task (i.e. lower decision bound needed in order to achieve target level of task performance) really were also better at metacognitive evaluation as well. Even in the within-subject design of Expt 1, it could be that the different mental set adopted in the "speed" condition induced changes that manifested partially as real differences in metacognitive ability. The correlations alone are insufficient to determine which interpretation is more likely to be correct. Thus, the entire rationale for interpreting the M_ratio correlations as spurious comes down to contrasting these with the non-significant correlations of v_ratio with decision bound. However, these v_ratio results themselves depend on a simplified modeling approach that only considers median confidence RT rather than full confidence RT distributions, and more crucially, depend on conf RT data that are likely heavily influenced by a large motor latency/noise component that may (1) obscure the decisional component of conf RT, (2) obscure the relationship between conf RT and choice RT. This issue with the quality of the conf RT data then casts doubt on the v_ratio fits, which in turn casts doubt on the interpretation of the M_ratio results, and ultimately casts doubt on the main argument of the paper. The authors propose to address these points about confidence RT in future modeling work, but in the absence of addressing them satisfactorily for the *present* work, it cannot stand as a really strongly held together scientific argument and instead comes off as an intriguing idea that is still in need of strong empirical support and (ideally) further sophistication in the modeling. Characterizing drift rate well requires characterizing the corresponding RTs well, so measuring and modeling confidence RT is something the arguments of this paper can't afford not to get right if it's going to be a really strong and compelling work.*

We appreciate the Reviewer's request for more scrutiny, and therefore we decided to replicate the critical finding that instructions concerning the tradeoff between speed and accuracy affect M-ratio but not v-ratio (cf. Experiment 1 in the previous version of our manuscript). To this end, we collected data of 32 new participants performing the same experiment, with the only difference that now both the choice and the confidence judgments were indicated using (separate) discrete key presses. Thus, in this novel dataset we now do have a reliable measurement of confidence RTs. These novel data fully replicate the original finding: instructing participants to focus on speed vs accuracy influenced the estimated decision boundary, $t(31) = 5.59, p < .001$, and also affected the estimation of M-ratio, $t(31) = 2.29, p = .029$. Importantly, our novel measure of metacognitive accuracy, v-ratio, which controls for differences in response caution, did not differ between both instruction conditions, $t(31) = 0.46, p = .647$. In sum, these additional data again confirm that a widely used measure of metacognitive accuracy, M-ratio, is affected by differences in decision boundary, whereas our novel dynamic measure of metacognitive accuracy is not. In the revised version of the manuscript, we have relegated the old Experiment 1 to the Supplementary Materials and in the main text now report this novel data as Experiment 1. We hope that the Reviewer agrees with us that these novel data, with more precise

measurement of confidence RTs, together with the updated model fitting procedure which takes into account the actual distributions of confidence RTs, satisfactorily address the remaining concerns that the Reviewer expressed.

Other points

** Model simulation: RT correlations and M_ratio*

*I appreciate the authors' revision of the model simulations so as to roughly reproduce the correlation coefficients between choice RT and conf RT seen in Expts 1-3. However, this only partially addresses the issue, since correlation does not take scaling into account. (e.g. the correlation between X and Y is the same as the correlation between 10*X and 100*Y.) The practical concern here is that even if the simulated choice RT and conf RT have similar correlation to what is seen in the empirical data, this doesn't ensure that the actual *magnitudes* of the conf RTs in relation to the choice RTs will reflect the empirical patterns. It appears as though the simulated choice and conf RTs do in fact have roughly similar relative magnitudes as those found in the data, but this was not highlighted in the relevant discussion and is not entailed by the similar correlation coefficient alone. For instance, had all the simulated conf RTs been multiplied by 100, that would still yield the same correlation coefficient between simulated choice RT and conf RT, even though it obviously would not be a good reflection of the data.*

We thank the Reviewer for making this point. In the revised version of our manuscript, we no longer simulated post-decision processing time using a fixed value, but instead simulated these using full RT distributions (in line with the updated modeling framework). In order to get confidence RT distributions, we first generated data which were sampled using a boundary and a drift rate both of which were sampled from a normal distribution ($\sigma = 1$) around the true bound and drift for that simulated observer, respectively. To account for the empirical observation that confidence RTs are usually faster than choice RTs, we fixed non-decision time to zero for these simulations. Finally, during the simulations, post-decision processing times were sampled from these “dummy” distributions. This procedure induced a moderate correlation between choice RTs and confidence RTs, $r(98) = .54, p < .001$, as seen in the data. Full details about this procedure are described on p. 19. We acknowledge that some of the choices made seem arbitrary, but we do want to stress that the simulation results are very robust to such choices. Specifically, we reran the simulations with different levels of noise for the dummy data, and with the restriction that drift rates were fixed across participants. In all these cases, the simulations consistently showed the same pattern of results. We therefore added the following on p. 18:

“Note that the simulation results were very robust, as the same findings were obtained when for example increasing or decreasing the noise for generating dummy data, when restricting post-decision processing time to a fixed value, or when only using a single drift rate for all simulations.”

Nevertheless, if the Reviewer would like to see another approach concerning these simulations, we are happy to add these to the manuscript.

A more substantive point about the simulated data is that many simulated data points have implausibly high M_ratio values. M_ratio is typically observed to be close to 1, roughly in line with theoretical expectation. Values greater than 1 do occur empirically, but values above 2 are almost never seen except in outlier cases where accurate estimation becomes difficult (e.g. due to low N or very low values of d'). This is born out in Expt 1-3, where M_ratio tops out at about 1-2. Yet the simulations involve many unrealistic cases where M_ratio ranges from 2-4. It appears that inclusion of these extreme M_ratio values artificially influences the correlation with both simulated v_ratio and simulated decision bound (Fig 1C). The authors should choose simulation parameters that do not yield M_ratio values above 2, and ideally have the majority of cases closer to or below M_ratio = 1, more in line with the range of M_ratio values typically observed in real data that does not suffer from statistical estimation issues.

As requested, in the revised version of the manuscript we have now chosen the parameters for the simulations in a way that assures that M-ratio values are within the range suggested by the Reviewer.

Note that the change made to the simulations reported above (i.e., simulating entire confidence RT distributions) had a strong impact on reducing the range of M-ratio.

** rating confidence from $p(\text{correct})$*

*The authors have nicely shown that computing confidence from $p(\text{correct} | e, t+s, X)$ does not entail that area under the type 2 ROC > 0.5 when $v_ratio=0$, or when post-decision accumulation time=0. However, they show that this is only the case when simulating data with a single drift rate, and does not hold when simulating with two drift rates. I suspect that the reason for this pattern of results is that, in DDMs where drift rate is constant, the RT distributions for correct and incorrect responses are the same-- which fails to capture the common empirical pattern whereby correct responses have faster RTs. This RT pattern can be captured, however, by introducing trial-by-trial variability in drift rate (Ratcliff & Rouder, 1998). Thus, in the authors' simulations with a single (non-varying) drift rate, RT distributions for correct and incorrect responses are likely the same, entailing that estimation of $p(\text{correct})$ carries no useful information when $v_ratio=0$ or post-decision accumulation time=0-- hence, type 2 AUC = 0.5. Whereas simulating two drift rates introduce some drift rate variability, and therefore differences in RT distributions for correct and incorrect trials that can be used to diagnose accuracy to some extent even when $v_ratio=0$ or post-decision accumulation time=0. Thus, I think the point raised in my original review still stands-- there is a conceptual tension between (1) estimating confidence from $p(\text{correct})$ in the way the authors do, and (2) characterizing metacognitive accuracy using v_ratio , since on this formulation v_ratio does not contain all the relevant information entering into confidence ratings. The authors' demonstration that type 2 AUC=0.5 when $v_ratio=0$ or post-decision accumulation time=0 thus only reflects the fact that their current modeling choices cannot account for differences in RT for correct and incorrect trials, which is a shortcoming. A fuller model implementation that could account for such RT differences might well exhibit type 2 AUC appreciably above-chance even when $v_ratio=0$ or post-decision accumulation time=0. (Such a model would presumably have more drift variance than the authors' two drift rate simulation, and therefore might have type 2 AUCs appreciably larger than the 0.511 value found for the two drift rate simulation.) This would then reintroduce the conceptual tension that v_ratio is not measuring everything there is to metacognitive accuracy after all, and at bottom the real work is being done by the unexplained $p(\text{correct})$ calculation. The authors write, "Finally, we feel that, rather than it being our aim to dissociate between these two very similar models, the goal of the current work is to demonstrate that static models of metacognition are too simplistic, and that instead dynamic models should be used." I take the point that the $p(\text{correct})$ issue is not the most central issue to address, but it is one that still seems in need of addressing nonetheless. And it is not a matter of dissociating the $p(\text{correct})$ version of the model from the more purely Pleskac & Busemeyer-type version of the model, so much as it is a matter of choice and conceptual interpretation. If the authors want to characterize v_ratio as a *complete* measure of metacognitive accuracy, this does not seem to leave room for the $p(\text{correct})$ implementation of the model which allows for influences on metacognitive accuracy outside of v_ratio . Conversely, if the authors want to use the $p(\text{correct})$ version of the model, this does not seem to leave room for interpreting v_ratio as a complete measure of metacognitive accuracy. I think either choice is viable, but just want to point out that the authors *do* have a choice to make here and can't have their cake and eat it too.*

As suggested by the Reviewer, we have updated our modeling framework and now model confidence directly as the strength of post-decisional evidence, as previously done by Pleskac & Busemeyer (2010, *Psych Rev*). This approach effectively solves the issue about type 2 AUC, which is at chance level with post-decision accumulation time equaling 0 and/or with v -ratio equaling 0. As the Reviewer indicates, this implementation allows us to characterize v -ratio as a complete measure of metacognitive accuracy, which is the main aim of the current work. In the Discussion, we have amended the previously reported text about Type-II ROC curves and instead explain why it is important to use the current implementation. The relevant paragraph can be found on p. 17:

‘Importantly, the choice to model decision confidence as a function of post-decisional evidence was directly informed by this finding. An alternative approach in the literature within the context of evidence accumulation models has been to quantify confidence as the probability of being correct

given time, evidence and the response made^{33,54-57}. Although this notion of decision confidence has been very successful in explaining empirical patterns seen in the literature, one drawback of this approach is that it does not predict chance-level type-II ROC performance with post-decision drift rate equal to zero, in the case of multiple drift rates. The reason that the probabilistic confidence model can still dissociate corrects from errors in this situation, is because it infers probability correct based on decision times (and both probability correct and decision times co-vary with drift rates). We therefore decided to quantify decision confidence as a function of post-decisional evidence, which allows to characterize v-ratio as a complete an unbiased measure of metacognitive accuracy.”

And in the Supplementary Materials, we have replaced the previous type II ROC simulations with those of the novel model simulations.

REVIEWER COMMENTS

Reviewer # 2 (Remarks to the Author):

The authors have addressed my last remaining concerns. The addition of the new data in particular have offered additional clarity. I've also had a look at the responses to R3 and in particular the extension of the model to capture the full distribution of confidence RTs and it is reassuring to see that the main conclusions of the study still stand. I am personally happy with the latest version of the manuscript.

Reviewer # 3 (Remarks to the Author):

My main concern with the previous version of the manuscript was the modeling of confidence RTs. The authors employed a diffusion model that assumed a constant post-decision accumulation time, which cannot account for variability in the confidence RT distribution. This is significant because in diffusion models, fitted drift rate (in conjunction with fitted decision bounds) is sensitive to both accuracy and RT data, and the authors' main results pertain to the fitting of post-decision drift rate. Thus, it is possible that in an expanded diffusion model that captures full confidence RT distributions, the fitting results for v_{ratio} might also change, which could affect the central results and interpretation of the manuscript.

In their response letter, the authors stated that "we further improved our modelling efforts and now explicitly fitted our computational model to the entire confidence RT distributions (instead of a single summary metric)." However, such a change is not reflected in the Methods section. The "Fitting procedure" section of the Methods appears to be largely identical to the corresponding section in the previous version of the manuscript, with the exception of changes to eq. (3) and the corresponding discussion. (This edit to the Methods addresses another point in my last review regarding how confidence is computed in the model, but is separate from the issue regarding the fitting of confidence RT distributions.)

In particular, in the most recent version of the manuscript, in the Methods it is still stated that "Specifically, for each participant we calculated the difference in time between the moment that participants made their initial choice and the moment that they confirmed their confidence judgment. From these differences we calculated, per participant, the median and used this value as the duration of post-decision processing time." This is the same method used in the previous version of the manuscript which raised my concern about fitting confidence RTs. The discrepancy between the response letter and the manuscript is not limited to the Methods but occurs elsewhere as well, e.g. in a description of the model in the Results section which implies the model had constant confidence RT and thus did not fit confidence RT distributions: "Afterwards, evidence continued to accumulate for a specified amount of time." (lines 190-191)

Additionally, there is no new text in the Methods or Results discussing the fitting of confidence RT distributions. There is also no data regarding the fits of the model to confidence RT distributions in the main manuscript or supplementary information.

Thus, the crucial change to the model regarding the fitting of confidence RT distributions that the authors refer to in the response letter is not in fact reflected in the revised manuscript. It is not clear to me if relevant edits to the Methods and other parts of the manuscript were accidentally omitted (in which case, the existing Methods and related discussions in Results and elsewhere are inaccurate and crucial information about confidence RT fitting methods and results are missing), or if the details discussed in the Methods and elsewhere are in fact accurate (in which case, the response letter does not accurately represent the revised manuscript, and the manuscript does not satisfactorily resolve a crucial issue). Unfortunately, this ambiguity precludes me from being able to assess if the revision in its current form satisfactorily addresses the points raised in the last review.

Reviewer #2 (Remarks to the Author):

The authors have addressed my last remaining concerns. The addition of the new data in particular have offered additional clarity. I've also had a look at the responses to R3 and in particular the extension of the model to capture the full distribution of confidence RTs and it is reassuring to see that the main conclusions of the study still stand. I am personally happy with the latest version of the manuscript.

We would like to thank Reviewer 2 for their appreciation of our previous revision.

Reviewer #3 (Remarks to the Author):

My main concern with the previous version of the manuscript was the modeling of confidence RTs. The authors employed a diffusion model that assumed a constant post-decision accumulation time, which cannot account for variability in the confidence RT distribution. This is significant because in diffusion models, fitted drift rate (in conjunction with fitted decision bounds) is sensitive to both accuracy and RT data, and the authors' main results pertain to the fitting of post-decision drift rate. Thus, it is possible that in an expanded diffusion model that captures full confidence RT distributions, the fitting results for v_{ratio} might also change, which could affect the central results and interpretation of the manuscript.

In their response letter, the authors stated that "we further improved our modelling efforts and now explicitly fitted our computational model to the entire confidence RT distributions (instead of a single summary metric)." However, such a change is not reflected in the Methods section. The "Fitting procedure" section of the Methods appears to be largely identical to the corresponding section in the previous version of the manuscript, with the exception of changes to eq. (3) and the corresponding discussion. (This edit to the Methods addresses another point in my last review regarding how confidence is computed in the model, but is separate from the issue regarding the fitting of confidence RT distributions.)

In particular, in the most recent version of the manuscript, in the Methods it is still stated that "Specifically, for each participant we calculated the difference in time between the moment that participants made their initial choice and the moment that they confirmed their confidence judgment. From these differences we calculated, per participant, the median and used this value as the duration of post-decision processing time." This is the same method used in the previous version of the manuscript which raised my concern about fitting confidence RTs. The discrepancy between the response letter and the manuscript is not limited to the Methods but occurs elsewhere as well, e.g. in a description of the model in the Results section which implies the model had constant confidence RT and thus did not fit confidence RT distributions: "Afterwards, evidence continued to accumulate for a specified amount of time." (lines 190-191)

The Reviewer is correct that we indeed failed to update the Methods section (and also other parts in the manuscript, e.g., lines 190-191) of the revised manuscript in light of the revisions to our model. We would like to apologize for the inconvenience this has caused the Reviewer. We did indeed adapt our model so that post-decision processing time no longer depends on a single value (which was the case in the previous version of the model), but instead we used the entire distribution of confidence RTs to directly determine the distribution of post-decision processing times. Importantly, this improvement in our model fit did not affect any of our conclusions.

We have now made this methodological improvement very explicit in the Results section, p.9 lines 191-192:

“The distribution of post-decision evidence accumulation times was directly determined by the distribution of empirically observed confidence RTs”

And more elaborated in the Methods, p.19, lines 452-459:

“After boundary crossing, the evidence continued to accumulate for a duration determined by the empirically observed confidence RT distribution (i.e., the difference in time between initial choice and confidence judgment). Specifically, the post-decision accumulation time of each simulated trial was set to be equal to the duration of a randomly selected trial from the confidence RT distribution of that participant. Note that this random selection was done without replacement, ensuring that the simulated confidence RT distribution exactly matched the empirically observed confidence RT distribution. Because the number of simulated trials always exceeded the number of empirical trials, sampling from the empirical confidence RT distribution restarted after all values were selected.”

Finally, on p. 17 and 18 we made a couple of minor changes (appearing in blue in the revised manuscript) to reassure that our modeling procedure was crystal clear to the reader.

Additionally, there is no new text in the Methods or Results discussing the fitting of confidence RT distributions. There is also no data regarding the fits of the model to confidence RT distributions in the main manuscript or supplementary information.

Note that we did not fit confidence RT distributions themselves, but instead we used confidence RT distributions to determine the distribution of post-decisional processing times. As explained above, for each simulated trial we allowed post-decisional processing to last for a period equal to that of a randomly selected trial from the confidence RTs distribution (without replacement). This approach effectively resolves the Reviewer’s concern that we did not take the full confidence RT distribution into account, while at the same time remaining agnostic regarding the specific stopping rule for confidence judgments (for which there seems no consensus yet in the literature).

Thus, the crucial change to the model regarding the fitting of confidence RT distributions that the authors refer to in the response letter is not in fact reflected in the revised manuscript. It is not clear to me if relevant edits to the Methods and other parts of the manuscript were accidentally omitted (in which case, the existing Methods and related discussions in Results and elsewhere are inaccurate and crucial information about confidence RT fitting methods and results are missing), or if the details discussed in the Methods and elsewhere are in fact accurate (in which case, the response letter does not accurately represent the revised manuscript, and the manuscript does not satisfactorily resolve a crucial issue). Unfortunately, this ambiguity precludes me from being able to assess if the revision in its current form satisfactorily addresses the points raised in the last review.

Again, we wish to offer our apologies for not including this information into the earlier revised version of our manuscript. Of course, the new version of the manuscript corrects this oversight and describes the changes to our fitting procedure in detail.

REVIEWER COMMENTS

Reviewer # 3 (Remarks to the Author):

In my previous review I pointed out that a discrepancy between the claims of the authors' reply letter and the contents of the revised manuscript precluded me from evaluating the revision fully. The authors have now satisfactorily addressed this discrepancy by updating the manuscript to specify how the modeling takes into account RT distributions for confidence ratings, in a way that is consistent with what was described in their reply letter. With this correction it is now possible for me to more fully consider and respond to the authors' previous reply letter. That reply letter addressed four main points: (1) fitting the model to full confidence RT distributions; (2) concerns about the quality of the empirical confidence RT data; (3) the computation of confidence in the model; and (4) the modeling of empirical correlations between choice RT and confidence RT. Below I consider the authors' treatment of these four points.

1 . fitting the model to full confidence RT distributions

The authors incorporate the empirical confidence RT distribution into their model fitting by allowing post-decision evidence accumulation to continue for N time steps on a given trial, where N is determined by sampling from the empirical distribution of confidence RTs. This approach ensures that the model produces a full distribution of confidence RTs in line with the empirical distribution, which allays concerns that failure to do so could muddy interpretation of the modeling results for v_{ratio} .

For present purposes I think this approach is sufficient, i.e. it is enough to show that when the DDM is constrained to reproduce empirical confidence RT distributions, fitted v_{ratio} is independent of fitted decision bound.

However, I also note that the specific manner in which this method captures the confidence RT distributions in the modeling results is ad hoc-- the model does not generate the confidence RT distributions in a manner analogous to how the classical DDM generates distributions of choice RT from the interaction of drift rate and decision bound, but rather this behavior is imposed on the model in kind of a brute force way. One can imagine using a similar method for fitting drift rate to choice RT in a modified version of the DDM, i.e. dispensing with the decision bound and finding the drift rate that reproduces accuracy data, given that the drift rates are artificially constrained to reproduce empirical choice RT distributions by an RT sampling procedure. Such a model would fit choice and RT data but would be greatly impoverished in the insight it could provide into psychological processes underlying choice behavior, as compared to the insights afforded by classical DDMs. In a similar way, the current instantiation of the v_{ratio} model remains limited in how much insight it provides for understanding confidence rating behavior in a DDM context due to the manner in which it imposes confidence RT distributions, rather than showing how these arise organically from the processes characterized by the model.

The issue of the stopping rule for the second stage of accumulation in DDMs of confidence is very much an open research question, as the authors note. However, this question is perhaps more salient for the v_{ratio} modeling framework than previous DDM approaches to modeling confidence, since unlike previous models it proposes to fit a separate drift rate for the second stage of evidence accumulation. In turn the fitting of this second-stage drift rate naturally invites a complete characterization of confidence accuracy and RT behavior that emerges organically from the model dynamics, in a way analogous to the DDM treatment of choice and RT. The current iteration of the v_{ratio} model, though it marks some interesting modeling innovations, is not there yet. This is not something that needs to be resolved in the current manuscript, but it would be appropriate for the authors to acknowledge this issue in the Discussion and identify it as an area for development in future research.

2 . concerns about the quality of the empirical confidence RT data

Previously, experiments 1-3 all had problematic experimental designs for the research question being investigated. This is because choice was entered via a single keypress, whereas confidence

was entered either by clicking a mouse on a continuous scale (expt 1) or by navigating a cursor along a continuous scale using keypresses (it is not clear if subjects could hold down a key to continuously move the cursor, or if they had to press keys multiple times to incrementally move the cursor-- this should be specified in the Methods), and then using an additional keypress of the "enter" key to officially register the confidence rating. In both cases, the motoric processes used to enter confidence ratings are very different from those used to enter choice-- likely slower and more variable. In expts 2-3, an additional difficulty is imposed by the fact that the method for confidence entry likely introduced a confound between confidence rating and confidence RT-- assuming constant cursor speed, it takes more time for the cursor to move to more extreme values on the confidence scale, thus artificially distorting the relationship between confidence rating and confidence RT.

As I described in a previous review, this is a crucial issue for the current research question, since v_{ratio} hinges on the comparison of the fitted pre- and post-decision drift rate. If the motoric processes used to generate choice and confidence are very different in their latency and variability, then the empirical RTs for choice and confidence are not fully comparable as psychological quantities-- the motor confound makes direct comparison impossible. It is a strong possibility that this difference in the motor component of the RTs could influence the fitted pre- and post-choice drift rates and hence v_{ratio} . In particular, the fitted post-choice drift rate would "absorb" any increase in latency and variability due to slower, non-single-keypress entry of confidence ratings. This could act as a source of bias and noise in the fitted post-choice drift rates which could in turn obscure relationships between v_{ratio} and other quantities, such as the fitted decision bound. Thus, it is possible that a key part of the authors' argument-- a null effect in the relationship between fitted v_{ratio} and decision bound in empirical data-- may be influenced by a motor confound that would add noise to the data and tend to increase the likelihood of finding spurious null effects.

The authors partially addressed this point by running a revised version of expt 1 using a single keypress for entry of the confidence rating, rather than the mouse click on a continuous scale, and moving the previously labelled expt 1 to supplementary material. This treatment satisfactorily addresses the motor confound issue for expt 1, and I applaud and commend the authors' efforts in revising and re-running this experiment.

However, unfortunately the experimental design for expts 2 and 3 remains problematic. The authors have not revised and re-run an improved version of these experiments or even acknowledged and defended against their shortcomings in the Discussion. In my view these experiments remain poorly suited to the primary research question at hand, and provide only weak and suggestive support for the conclusions drawn from the null findings in those experiments. It remains a distinct possibility that the null effect in these experiments that is central to the authors' conclusions is artificially generated by the motor confound introduced by the experimental design. Thus a core component of the results and interpretation of the paper remains problematic in its current form.

Notably, while it is reassuring that the revised expt 1 produced similar results as the original expt 1, this is no guarantee that a revised expt 2 would produce similar results to the original expts 2 and 3. Important differences are that (1) expt 2 explores the relationship of spontaneously utilized decision bound and post-decision drift rate, whereas expt 1 manipulates decision bound with task instructions; (2) the original expt 1 and expt 2 used different methods for confidence entry, meaning that the nature of the motor confound introduced could also differ between these experiments.

The best way to address this issue would be to revise and re-run expt 2 in the same manner as the authors did for expt 1. This approach would ensure that the data, modeling, and conclusions for the revised experiment are of sufficiently high quality to provide strong support for the authors' arguments, assuming that a revised data set continued to show no relationship between spontaneous decision bounds and post-choice drift rate.

However, I also recognize that the authors have already done a lot of work to revise and improve the manuscript up to this point, and I do not wish to protract this unusually long review process

even further by making a revised experiment a make-or-break request here. So if the authors do not have the time or resources to run another experiment, I think it is sufficient to leave the existing expts 2-3 as-is provided that the issues described above are given sufficient attention and development in the Discussion.

Should the authors wish to go this route rather than revise and re-run expt 2, this added part of the Discussion should accomplish the following:

- (1) address the general importance for the v -ratio framework of ensuring that choice and confidence RTs be as comparable as possible for the purposes of making the fitted pre- and post-choice drift rates as comparable as possible;
- (2) address the role of best practices for experimental design in data sets to be analyzed within a v -ratio framework, e.g. ensuring that the motoric processes used to enter choice and confidence are as similar as possible to prevent large differences in the motor component of RTs from muddying the pre- and post-choice data and therefore v -ratio itself, plus any other factors the authors can think of here;
- (3) acknowledge that in light of all this, expts 2-3 do not feature ideal design for v -ratio analysis for all the reasons discussed above, which renders the results and interpretation of expts 2-3 as relatively weak and suggestive evidence that spontaneous decision bound and post-choice drift rate are uncorrelated, and that this finding is therefore in need of further confirmation in future research using more appropriate experimental design.

3 . the computation of confidence in the model

The authors have satisfactorily addressed the points in my previous review regarding how confidence is computed by virtue of revising the Methods and also providing helpful further comments on the rationale for this modeling choice in the Discussion.

4 . the modeling of empirical correlations between choice RT and confidence RT

This is a minor point, but the revised text in the manuscript that the authors wrote in response to a previous point about choice and confidence RT correlation is rather unclear to me. The relevant text is on lines 432 - 442 of page 18. Lines 435 - 438 make it sound like the authors are using some random sampling of parameters in the model that govern choice behavior, but the connection of this to the confidence behavior of the model is very opaque. And on line 440 it is unclear what the "dummy" RT distributions are. It seems like this is in reference to the empirical distribution of confidence RT in the data, but if so, why is it called a "dummy" distribution?

Reviewer #3

In my previous review I pointed out that a discrepancy between the claims of the authors' reply letter and the contents of the revised manuscript precluded me from evaluating the revision fully. The authors have now satisfactorily addressed this discrepancy by updating the manuscript to specify how the modeling takes into account RT distributions for confidence ratings, in a way that is consistent with what was described in their reply letter. With this correction it is now possible for me to more fully consider and respond to the authors' previous reply letter. That reply letter addressed four main points: (1) fitting the model to full confidence RT distributions; (2) concerns about the quality of the empirical confidence RT data; (3) the computation of confidence in the model; and (4) the modeling of empirical correlations between choice RT and confidence RT. Below I consider the authors' treatment of these four points.

We thank the Reviewer again for their careful consideration of our reply.

1. fitting the model to full confidence RT distributions

The authors incorporate the empirical confidence RT distribution into their model fitting by allowing post-decision evidence accumulation to continue for N time steps on a given trial, where N is determined by sampling from the empirical distribution of confidence RTs. This approach ensures that the model produces a full distribution of confidence RTs in line with the empirical distribution, which allays concerns that failure to do so could muddy interpretation of the modeling results for v_ratio .

For present purposes I think this approach is sufficient, i.e. it is enough to show that when the DDM is constrained to reproduce empirical confidence RT distributions, fitted v_ratio is independent of fitted decision bound.

However, I also note that the specific manner in which this method captures the confidence RT distributions in the modeling results is ad hoc-- the model does not generate the confidence RT distributions in a manner analogous to how the classical DDM generates distributions of choice RT from the interaction of drift rate and decision bound, but rather this behavior is imposed on the model in kind of a brute force way. One can imagine using a similar method for fitting drift rate to choice RT in a modified version of the DDM, i.e. dispensing with the decision bound and finding the drift rate that reproduces accuracy data, given that the drift rates are artificially constrained to reproduce empirical choice RT distributions by an RT sampling procedure. Such a model would fit choice and RT data but would be greatly impoverished in the insight it could provide into psychological processes underlying choice behavior, as compared to the insights afforded by classical DDMs. In a similar way, the current instantiation of the v_ratio model remains limited in how much insight it provides for understanding confidence rating behavior in a DDM context due to the manner in which it imposes confidence RT distributions, rather than showing how these arise organically from the processes characterized by the model.

The issue of the stopping rule for the second stage of accumulation in DDMs of confidence is very much an open research question, as the authors note. However, this question is perhaps more salient for the v_ratio modeling framework than previous DDM approaches to modeling confidence, since unlike previous models it proposes to fit a separate drift rate for the second stage of evidence accumulation. In turn the fitting of this second-stage drift rate naturally invites a complete characterization of confidence accuracy and RT behavior that emerges organically from the model dynamics, in a way analogous to the DDM treatment of choice and RT. The current iteration of the v_ratio model, though it marks some interesting modeling innovations, is not there yet. This is not something that needs to be resolved in the current manuscript, but it would be appropriate for the authors to acknowledge this issue in the Discussion and identify it as an area for development in future research.

We thank the reviewer for this comment. We feel that our agnostic approach in estimating v -ratio is currently the preferred approach because it is constrained to reproduce empirical confidence RT

distributions. Therefore, in the Discussion we point out that indeed future work should attempt to further unravel the stopping rule underlying confidence judgments:

“Finally, we note that in the modeling efforts reported here, the duration of post-decisional evidence accumulation was decided based on the full distribution of empirically observed confidence RTs. Most previous modeling efforts have likewise assumed that post-decision processing terminates once confidence is externally queried²⁸, and only a few studies have explicitly examined different stopping rules for post-decision processing^{32,33}. Given that we still lack a clear mechanistic understanding of how post-decisional processing is terminated, we here decided for this implementation which is agnostic regarding the underlying stopping criterion for confidence judgments, but nevertheless takes the full distribution of confidence RTs into account during fitting. By further unravelling the computational mechanisms underlying post-decisional accumulation termination, substantial progress can still be made by including these mechanisms in future modeling efforts.”

2. concerns about the quality of the empirical confidence RT data

Previously, experiments 1-3 all had problematic experimental designs for the research question being investigated. This is because choice was entered via a single keypress, whereas confidence was entered either by clicking a mouse on a continuous scale (expt 1) or by navigating a cursor along a continuous scale using keypresses (it is not clear if subjects could hold down a key to continuously move the cursor, or if they had to press keys multiple times to incrementally move the cursor-- this should be specified in the Methods), and then using an additional keypress of the "enter" key to officially register the confidence rating. In both cases, the motoric processes used to enter confidence ratings are very different from those used to enter choice-- likely slower and more variable. In expts 2-3, an additional difficulty is imposed by the fact that the method for confidence entry likely introduced a confound between confidence rating and confidence RT-- assuming constant cursor speed, it takes more time for the cursor to move to more extreme values on the confidence scale, thus artificially distorting the relationship between confidence rating and confidence RT.

As I described in a previous review, this is a crucial issue for the current research question, since v_{ratio} hinges on the comparison of the fitted pre- and post-decision drift rate. If the motoric processes used to generate choice and confidence are very different in their latency and variability, then the empirical RTs for choice and confidence are not fully comparable as psychological quantities-- the motor confound makes direct comparison impossible. It is a strong possibility that this difference in the motor component of the RTs could influence the fitted pre- and post-choice drift rates and hence v_{ratio} . In particular, the fitted post-choice drift rate would "absorb" any increase in latency and variability due to slower, non-single-keypress entry of confidence ratings. This could act as a source of bias and noise in the fitted post-choice drift rates which could in turn obscure relationships between v_{ratio} and other quantities, such as the fitted decision bound. Thus, it is possible that a key part of the authors' argument-- a null effect in the relationship between fitted v_{ratio} and decision bound in empirical data-- may be influenced by a motor confound that would add noise to the data and tend to increase the likelihood of finding spurious null effects.

The authors partially addressed this point by running a revised version of expt 1 using a single keypress for entry of the confidence rating, rather than the mouse click on a continuous scale, and moving the previously labelled expt 1 to supplementary material. This treatment satisfactorily addresses the motor confound issue for expt 1, and I applaud and commend the authors' efforts in revising and re-running this experiment.

However, unfortunately the experimental design for expts 2 and 3 remains problematic. The authors have not revised and re-run an improved version of these experiments or even acknowledged and defended against their shortcomings in the Discussion. In my view these experiments remain poorly suited to the primary research question at hand, and provide only weak and suggestive support for the conclusions drawn from the null findings in those experiments. It remains a distinct possibility that the null effect in these experiments that is central to the authors' conclusions is artificially generated by the motor

confound introduced by the experimental design. Thus a core component of the results and interpretation of the paper remains problematic in its current form.

Notably, while it is reassuring that the revised expt 1 produced similar results as the original expt 1, this is no guarantee that a revised expt 2 would produce similar results to the original expts 2 and 3. Important differences are that (1) expt 2 explores the relationship of spontaneously utilized decision bound and post-decision drift rate, whereas expt 1 manipulates decision bound with task instructions; (2) the original expt 1 and expt 2 used different methods for confidence entry, meaning that the nature of the motor confound introduced could also differ between these experiments.

The best way to address this issue would be to revise and re-run expt 2 in the same manner as the authors did for expt 1. This approach would ensure that the data, modeling, and conclusions for the revised experiment are of sufficiently high quality to provide strong support for the authors' arguments, assuming that a revised data set continued to show no relationship between spontaneous decision bounds and post-choice drift rate.

However, I also recognize that the authors have already done a lot of work to revise and improve the manuscript up to this point, and I do not wish to protract this unusually long review process even further by making a revised experiment a make-or-break request here. So if the authors do not have the time or resources to run another experiment, I think it is sufficient to leave the existing expts 2-3 as-is provided that the issues described above are given sufficient attention and development in the Discussion.

As requested, we have reanalyzed an old dataset and collected a new dataset that directly tackle this issue. Both datasets used the exact same response lay-out as in Experiment 1 (i.e. binary choice by pressing 'c' or 'n' with the thumbs, followed by a 6-choice confidence judgments with the fingers of both hands). In addition to the two datasets with speed-accuracy tradeoffs, our manuscript now features four datasets without any manipulation, which allowed us to directly investigate whether response modality (i.e. providing confidence with button presses vs on a continuous scale) plays a role in our design. To analyze the four datasets in a comprehensive way, we performed hierarchical mixed effects modeling on the subject-by-subject model fitted parameters accounting for the dependency of datasets on the specific experimental design. We focused on the two outstanding questions: (1) is M-ratio related to the height of the decision boundary, irrespective of the way in which confidence judgments are given, and (2) is v-ratio related to M-ratio but not to decision boundary.

(1) The relation between M-ratio and decision boundary

In a first analysis, we tested whether M-ratio was predicted by the decision bound and, crucially, whether this effect interacted with the mode of responding (i.e. without or without motor confound). To achieve this, we fitted the following hierarchical mixed effects model to the data:

$$\text{M-ratio} \sim \text{decision bound} * \text{response mode} + (1 \mid \text{experiment})$$

where (1 | experiment) reflects the hierarchical clustering of subjects within experiments and * indicates that an interaction effect was estimated. As expected, we observed a significant effect of boundary on M-ratio, $b = -.200$, $t(424) = -2.43$, $p = .015$ (see Figure R1). Importantly, this effect did not interact with response mode, $p = .118$, nor was there a main effect of response mode, $p > .337$. Thus, this analysis demonstrates that there was clear evidence across datasets for a relation between M-ratio and decision boundary irrespective of the way in which confidence was measured. This analysis shows, again, that M-ratio is not a pure measure of metacognition, but is confounded with response caution.

Figure R1. The relation between M-ratio and decision boundary across the four different data sets, $b = -.200$, $p = .015$. Note: the regression line shows the estimate from the hierarchical model fit, the transparent bands show the 95% confidence interval. Each dot reflects one participant, with the color depending on the dataset.

(2) Is v-ratio a good measure of metacognition?

Second, we addressed the question whether v-ratio is a good measure of metacognition. To do so, we tested whether v-ratio is related to M-ratio, showing that both measures capture shared variance in metacognition, and whether v-ratio is unrelated to the decision boundary, suggesting that v-ratio is independent of response caution. Crucially, we tested whether this depends on the mode of responding (i.e. with or without the motor confound). The full model in which we predicted v-ratio by M-ratio, decision boundary, and response mode and all interactions between these variables could not be estimated because the predictors were too strongly correlated (Variance Inflation Factors > 35). This was the case for M-ratio and its interaction with decision boundary, so we estimated the following reduced model:

$$v\text{-ratio} \sim M\text{-ratio} + \text{decision bound} * \text{response mode} + (1 | \text{experiment})$$

As expected, this analyses showed a strong relation between M-ratio and v-ratio, $b = .31$, $t(424) = 5.548$, $p < .001$ (see Figure R2, left panel). Importantly, there was no main effect of decision boundary, $p = .233$, no main effect of response mode, $p = .212$, nor an interaction between boundary and response mode, $p = .166$. Note that the absence of a relation between v-ratio and decision boundary was unrelated to the presence of M-ratio, when dropping this term from the model this did not change the results, $ps > .350$. To further show that the relation between v-ratio and M-ratio did not depend on response mode, we further fitted the following model:

$$v\text{-ratio} \sim M\text{-ratio} * \text{response mode} + (1 | \text{experiment})$$

As expected, in this model we again found the relation between M-ratio and v-ratio, $p < .001$, but no main effect of response mode, $p = .518$, nor an interaction effect, $p = .186$ (see Figure R2, right panel). Jointly, these two analyses show that v-ratio is an appropriate measure of metacognition because it is related to M-ratio but unrelated to the decision boundary.

Figure R2. The relation between M -ratio and v -ratio, $b=.31$, $p<.001$ (left panel) and v -ratio and decision boundary, $b = .11$, $p=.233$ (right panel). Same convention as in Figure R1.

In addition, below we provide the individual results of the two new experiments.

In the first new experiment ($N=67$), on each trial participants decided whether the average color of each element was red or blue, by pressing 'c' or 'n' with the thumbs of both hands. After their choice, they indicated their level of confidence using the same setup as in Experiment 1. The experiment featured 4 levels of difficulty, so during the fitting we estimated 4 separate drift rates, which explained the data very well. Generally, the model captured the data very well.

Results showed that, although there was no relationship between M -ratio and v -ratio, $r = -.02$, we again replicated the negative relationship between M -ratio and decision boundary, $r = -.29$, $p = .014$, whereas there was no relation between v -ratio and the estimated decision boundary, $r = -.18$, $p = .14$.

In the second new experiment ($N=96$), on each trial participants decided which one of two squared on the screen had most dots, by pressing ‘c’ or ‘n’ with the thumbs of both hands. After their choice, they indicated their level of confidence using the same setup as in Experiment 1. Again, our computational model fitted the data very well.

The results show a strong relation between M-ratio and v-ratio, $r = .62, p < .001$, but there was no relation between M-ratio and the decision boundary, $r = .111, p = .281$, and a small but significant relation between v-ratio and the estimated decision boundary, $r = .222, p = .029$.

Thus, although there is some variation between individual studies, our hierarchical mixed effects analysis reported earlier showed that, when considering all data collected so far, there is no clear evidence for an influence of response modality on our findings. Instead, we found that across all datasets M-ratio was negatively related to the decision boundary and positively to v-ratio.

We have added the results of our hierarchical mixed effects analysis, as well as results from all individual experiments, to the manuscript on p. 10-13.

Should the authors wish to go this route rather than revise and re-run expt 2, this added part of the Discussion should accomplish the following:

(1) address the general importance for the v_ratio framework of ensuring that choice and confidence RTs be as comparable as possible for the purposes of making the fitted pre- and post-choice drift rates as comparable as possible;

(2) address the role of best practices for experimental design in data sets to be analyzed within a v_ratio framework, e.g. ensuring that the motoric processes used to enter choice and confidence are as similar as possible to prevent large differences in the motor component of RTs from muddying the pre- and post-choice data and therefore v_ratio itself, plus any other factors the authors can think of here;

(3) acknowledge that in light of all this, expts 2-3 do not feature ideal design for v_ratio analysis for all the reasons discussed above, which renders the results and interpretation of expts 2-3 as relatively weak and suggestive evidence that spontaneous decision bound and post-choice drift rate are uncorrelated, and that this finding is therefore in need of further confirmation in future research using more appropriate experimental design.

Finally, in the Discussion on p. 17 we now added the following text to alert readers about the importance of measuring confidence RTs in a precise way:

“An important caveat is that in order to measure v-ratio as accurately as possible, precise measurements of confidence reaction times are needed. This is an important concern, because in many metacognition experiments confidence is queried using approaches that do not provide precise measurements of confidence RTs. In fact, in several of the experiments reported in the current manuscript confidence was queried using a mouse or by moving a cursor along the scale using the keyboard arrows. Although the results of Experiment 1 and Experiments 2A-D did not depend on the mode of confidence responses, we strongly advise researchers interested in deploying v-ratio to collect data using a design that measures the timing of both choices and confidence in a very precise manner.”

3. the computation of confidence in the model

The authors have satisfactorily addressed the points in my previous review regarding how confidence is computed by virtue of revising the Methods and also providing helpful further comments on the rationale for this modeling choice in the Discussion.

Thank you.

4. the modeling of empirical correlations between choice RT and confidence RT

This is a minor point, but the revised text in the manuscript that the authors wrote in response to a previous point about choice and confidence RT correlation is rather unclear to me. The relevant text is on lines 432 - 442 of page 18. Lines 435 - 438 make it sound like the authors are using some random sampling of parameters in the model that govern choice behavior, but the connection of this to the confidence behavior of the model is very opaque. And on line 440 it is unclear what the "dummy" RT distributions are. It seems like this is in reference to the empirical distribution of confidence RT in the data, but if so, why is it called a "dummy" distribution?

To make this clearer, we have now written the following on p. 19:

“To achieve this, for each simulated observer we selected a value for boundary and drift rate from a normal distribution ($\sigma = 1$) around the true boundary and true drift for that simulated observer, respectively. Then, we simulated a confidence RT distribution using these two values (ter was set to 0 to account for the fact that confidence RTs are usually faster than choice RTs). During the actual simulations, post-decision processing times were sampled from this confidence RT distribution.”

REVIEWERS' COMMENTS

Reviewer # 2 (Remarks to the Author):

I was asked to comment on the authors responses to R3's latest comments. On balance, I believe the authors have done a good job addressing the reviewer's comments, including collecting additional data and running further analyses. The original conclusions of the paper still stand and the discussion has been extended to further touch on the implications (and potential shortcomings) of the current work. In my opinion the manuscript can now be accepted. Ultimately, this will allow the rest of the community to further explore the utility and and/or develop extensions of the proposed framework.

Reviewer # 3 (Remarks to the Author):

I commend the authors for a very strong and thorough response, which I think has improved the quality of the manuscript significantly. All points from my previous review have been satisfactorily addressed.

There is one further general conceptual point that I think the authors would do well to touch on in the Discussion, which is that the critique of SDT-based analysis of type 2 sensitivity (M_{ratio}) depending on decision bound applies just as well to SDT-based analyses of type 1 sensitivity (d'). That is, d' in SDT is taken as a measure of type 1 sensitivity, but in a DDM context d' also depends on response bound (raising the bound allows for more accumulation of evidence and thus higher d'). Thus the point the authors raise here is not specific to M_{ratio} per se but applies more broadly to the relationship between static (e.g. SDT) and dynamic (e.g. DDM) modeling frameworks overall, including at the type 1 level. This relationship has not made d' obsolete as a measure of type 1 sensitivity, but rather the choice of whether SDT or DDM analysis is most appropriate for modeling type 1 data depends on the research question, experimental design, and other contextual factors. The same observation would seem to hold for the choice of whether to use M_{ratio} or v_{ratio} , as a specific instance of this more general pattern.

Including some text in the Discussion on this point would help give a broader context for the current work. I do not think it is necessary for there to be another round of review for this purpose; I trust whatever amendments the authors make will be satisfactory. I congratulate the authors on a very nice piece of work.

Reviewer #2 (Remarks to the Author):

I was asked to comment on the authors responses to R3's latest comments. On balance, I believe the authors have done a good job addressing the reviewer's comments, including collecting additional data and running further analyses. The original conclusions of the paper still stand and the discussion has been extended to further touch on the implications (and potential shortcomings) of the current work. In my opinion the manuscript can now be accepted. Ultimately, this will allow the rest of the community to further explore the utility and and/or develop extensions of the proposed framework.

We thank the Reviewer for appreciating our final revisions.

Reviewer #3 (Remarks to the Author):

I commend the authors for a very strong and thorough response, which I think has improved the quality of the manuscript significantly. All points from my previous review have been satisfactorily addressed.

We thank the Reviewer for appreciating our final revisions.

There is one further general conceptual point that I think the authors would do well to touch on in the Discussion, which is that the critique of SDT-based analysis of type 2 sensitivity (M_{ratio}) depending on decision bound applies just as well to SDT-based analyses of type 1 sensitivity (d'). That is, d' in SDT is taken as a measure of type 1 sensitivity, but in a DDM context d' also depends on response bound (raising the bound allows for more accumulation of evidence and thus higher d'). Thus the point the authors raise here is not specific to M_{ratio} per se but applies more broadly to the relationship between static (e.g. SDT) and dynamic (e.g. DDM) modeling frameworks overall, including at the type 1 level. This relationship has not made d' obsolete as a measure of type 1 sensitivity, but rather the choice of whether SDT or DDM analysis is most appropriate for modeling type 1 data depends on the research question, experimental design, and other contextual factors. The same observation would seem to hold for the choice of whether to use M_{ratio} or v_{ratio} , as a specific instance of this more general pattern. Including some text in the Discussion on this point would help give a broader context for the current work.

As suggested, we have added the following in the discussion on p. 15, lines 352-359:

“Notably, our claim that signal-detection theoretic measures of performance are confounded by response caution applies to both second-order performance measures (e.g., meta- d') as well as first-order performance measures (e.g. d'). Everything else being equal, lower decision boundaries will lead to lower values of d' , because choices will be made with less accumulated evidence. This knowledge, however, does not make the use of signal-detection theoretic measures obsolete; indeed, its usefulness depends on the research question, experimental design, and other contextual factors. Likewise, the choice for M_{ratio} versus v_{ratio} as a measure of metacognitive accuracy might depend on similar considerations.”

I do not think it is necessary for there to be another round of review for this purpose; I trust whatever amendments the authors make will be satisfactory. I congratulate the authors on a very nice piece of work.

Thanks.